# Epigenome-wide DNA methylation profiling in Progressive Supranuclear Palsy reveals major changes at *DLX1*

Axel Weber [1], Sigrid C. Schwarz[2,3], Jörg Tost [4], Dietrich Trümbach[5], Pia Winter[1], Florence Busato[4], Pawel Tacik[6,7], Anita C. Windhorst[8], Maud Fagny[4], Thomas Arzberger[3,9,10], Catriona McLean[11], John C. van Swieten[12], Johannes Schwarz[2], Daniela Vogt Weisenhorn[3,5,13], Wolfgang Wurst[3,5,13,14], Till Adhikary[15], Dennis W. Dickson [6], Günter U. Höglinger [2,3,14] & Ulrich Müller [1]

Genetic, epigenetic, and environmental factors contribute to the multifactorial disorder progressive supranuclear palsy (PSP). Here, we study epigenetic changes by genome-wide analysis of DNA from postmortem tissue of forebrains of patients and controls and detect significant ($P < 0.05$) methylation differences at 717 CpG sites in PSP vs. controls. Four-hundred fifty-one of these sites are associated with protein-coding genes. While differential methylation only affects a few sites in most genes, *DLX1* is hypermethylated at multiple sites. Expression of an antisense transcript of *DLX1*, *DLX1AS*, is reduced in PSP brains. The amount of DLX1 protein is increased in gray matter of PSP forebrains. Pathway analysis suggests that DLX1 influences *MAPT*-encoded Tau protein. In a cell system, overexpression of *DLX1* results in downregulation of *MAPT* while overexpression of *DLX1AS* causes upregulation of *MAPT*. Our observations suggest that altered *DLX1* methylation and expression contribute to pathogenesis of PSP by influencing *MAPT*.

[1] Institute of Human Genetics, Justus-Liebig-Universität, Gießen 35392, Germany. [2] Department of Neurology, Technische Universität München, Munich 81377, Germany. [3] German Center for Neurodegenerative Diseases (DZNE), Munich 81377, Germany. [4] Laboratory for Epigenetics and Environment, Centre National de Recherche en Génomique Humaine, CEA—Institut de Biologie Francois Jacob, Evry 91000, France. [5] Institute of Developmental Genetics, Helmholtz Center München, Munich 85764, Germany. [6] Department of Neuroscience, Mayo Clinic, Jacksonville, FL, 32224, USA. [7] Department of Neurodegenerative Diseases and Geriatric Psychiatry, University of Bonn Medical Center, Bonn 53127, Germany. [8] Institute of Medical Informatics, Justus-Liebig-Universität, Gießen 35392, Germany. [9] Department of Psychiatry, Ludwig-Maximilians-Universität, Munich 81377, Germany. [10] Center for Neuropathology and Prion Research, Ludwig-Maximilians-Universität, Munich 81377, Germany. [11] Alfred Anatomical Pathology and NNF, Victorian Brain Bank, Carlton, VIC, 3053, Australia. [12] Department of Neurology, Erasmus Medical Centre, Rotterdam 3000, The Netherlands. [13] Chair of Developmental Genetics, Technische Universität München-Weihenstephan, Neuherberg/Munich 85764, Germany. [14] Munich Cluster for Systems Neurology (SyNergy), Munich 81377, Germany. [15] Institute for Molecular Biology and Tumor Research, Center for Tumor Biology and Immunology, Philipps University, Marburg 35043, Germany. These authors contributed equally: Axel Weber, Sigrid C. Schwarz. These authors jointly supervised this work: Günter U. Höglinger, Ulrich Müller. Correspondence and requests for materials should be addressed to A.W. (email: axel.weber@humangenetik.med.uni-giessen.de) or to G.U.H. (email: Guenter.Hoeglinger@dzne.de) or to U.M. (email: Ulrich.mueller@med.uni-giessen.de)

Progressive supranuclear palsy (PSP) is a progressive and fatal neurodegenerative disease with a prevalence of 5−7/100,000[1,2]. Disease onset is usually beyond 60 years of age and the average survival time is 6–7 years after onset[1]. Symptoms include ocular motor dysfunction, postural instability, akinesia, cognitive dysfunction, and dysphagia, with the latter being the most frequent cause of death in PSP[1,2].

PSP is neuropathologically defined by intracellular aggregation of the microtubule-associated protein Tau in neurofibrillary tangles and tufted astrocytes. The aggregates eventually cause neuronal cell death in the cerebral cortex, diencephalon, brainstem, and cerebellar nuclei[3].

PSP is a "complex" disorder; genetic, environmental and epigenetic modifications contribute to disease. A variant of the gene MAPT is the major genetic risk factor in PSP[4,5]. Variants of the genes STX6, EIF2AK3, and MOBP also increase the risk of PSP[5]. Among environmental factors, advanced age is the best established risk factor[6]. Epigenetic modifications reported so far in PSP include aberrant DNA methylation at the MAPT locus[7–9] and miRNA dysregulation[10,11].

In order to learn more about the possible relevance of epigenetic changes in PSP we set out to study epigenetic alterations at the DNA level in prefrontal lobe tissue of PSP patients. We describe significant DNA methylation differences between patients and controls at many CpG sites, amounting to 451 protein-coding genes. While methylation differences only affect one or a few sites at most genes, highly significant ($\geq 5\%$) hypermethylation is found at multiple sites associated with the gene DLX1. Functional analyses of both DLX1 and its antisense transcript DLX1AS are consistent with an important role of DLX1 in the pathogenesis of PSP.

## Results

**Differentially methylated sites in PSP.** The genome-wide DNA methylation patterns of 94 PSP patients (72 ± 5.3 years; 57% male, 43% female) were compared to 71 controls (76 ± 7.9 years; 67% male, 33% female) without neurological or psychiatric diseases (Supplementary Data 1). We studied prefrontal lobe tissue since it is consistently pathologically damaged in PSP, but less so than other brain regions[3]. We estimated the amount of neuronal and non-neuronal cells in our samples, as described by Guintivano et al.[12]. The percentage of neurons in PSP patients (median 36.1% of cells) did not significantly differ from the proportion of neuronal cells in controls (median 38.0% of cells; Wilcoxon test, $P = 0.31$) (Supplementary Fig. 1). Thus, we detected disease-specific alterations and minimized a potential bias by a massive change in regional cellular composition due to severe neurodegeneration and gliosis.

Methylation differences at CpG sites between patients and controls were analyzed on 450 K BeadChips[13] applying a linear regression model with age, sex, and non-neuronal cell content as covariates at a Benjamini-Hochberg[14] corrected level of significance of $P < 0.05$. Significant CpGs previously shown to be influenced by genetic variants (mQTLs) in adult prefrontal cortex[15] were not included in further analyses. However, only three CpGs were found to match this criterion, i.e., cg01378667, cg03325535, cg10318222 in the promoter region of GABRA5, all of which are associated in cis with rs7496866 located in the same region. An influence of presently unknown genetic variants on the methylation pattern cannot be excluded. It is also not possible to correct for environmental factors such as individual medications and/or accompanying neurological diseases that might affect DNA methylation in the forebrain.

After these corrections, significant methylation differences were detected at 717 sites (627 hyper-, 90 hypomethylated). Mean differences of $\geq 5\%$ were found at 38 of these sites (34 hyper-, 4 hypomethylated) (Fig. 1a and Supplementary Data 2). Of the hypomethylated sites, 70% were associated with protein-coding genes, 4% with genes for non-coding RNAs (miRNAs, lncRNAs, etc.), and 26% were located beyond 1.5 kb of genes. The respective percentages for hypermethylated genes were 62, 3.5, and 34.5% (Fig. 1b).

The percentage of hypomethylated CpG sites within gene bodies was 40% for hypo- and 28% for hypermethylated sites (Fig. 1c). Twelve percent of hypo- and 8% of hypermethylated CpG sites were located within 5′UTRs (Fig. 1c). Seven percent hypo- and 2% hypermethylated sites were within the first exons of the respective genes.

Among sites hypomethylated in patients 44% were CpG islands and among hypermethylated sites 25% represented CpG islands. Thirty-four percent CpG sites were isolated ("open sea"[13]) and 31% of hypermethylated sites were in "open sea" regions. The percentages of hypomethylated CpG sites in "shelves" (2–4 kb from CpG island) was 6%, and that in "shores" (up to 2 kb from a CpG island[13]) was 17%. The corresponding percentages for hypermethylated sites are 8 and 36% (Fig. 1d).

**Chromosomal location of differentially methylated genes.** Figure 2 depicts the 375 genes with significant methylation differences between patients and controls and highlights the ten genes with differences $\geq 5\%$ (see also Supplementary Data 2). Eight of these genes are hyper- and two are hypomethylated.

In order to check validity of the BeadChip-based primary results we analyzed the methylation status at selected loci by pyrosequencing of bisulfite-converted DNA of the same samples. This confirmed the methylation differences in a representative subset of six genes, i.e., DLX1/DLX1AS, DLX2, METAP1D, SLC15A3, SLIT1, and TRRAP (Fig. 3c and Supplementary Fig. 2).

**No significant differential methylation of MAPT.** Analysis of the region 17q21.31 that spans 1.6 Mb (43,470,000 to 45,061,000 on chromosome 17) and includes MAPT revealed nominally significant methylation differences at 11 sites. However, the findings were not significant after correction for multiple testing. The smallest $P$-value corrected for multiple testing within MAPT was 0.0578 at chr17:44026659 (Supplementary Data 3).

**Pronounced hypermethylation of DLX1.** Differential methylation of $\geq 5\%$ was only detected at a few CpG sites in a small number of genes in PSP (Fig. 2 and Supplementary Data 2). Most pronounced hypermethylation was detected at a region of chromosome 2 that includes the gene DLX1 (Distal-Less Homeobox 1). Many sites of DLX1, mainly within its 3′UTR, were hypermethylated by $\geq 5\%$, as shown in a representative heat-map of 11 sites (Fig. 3a).

The DLX1 gene is composed of three exons[16]. Greatest methylation differences were found at a CpG island (i.e., a genomic region of > 200 bp with a CG content of > 50% and an observed/expected CpG ratio of > 60%) in the 3′UTR, spanning positions 172952810–172953160 [hg19] on chromosome 2 (Fig. 3b). Pyrosequencing confirmed hypermethylation of nine CpGs within the CpG island that is located in the 3′UTR of DLX1 (Fig. 3c).

**DLX1 transcript.** We proceeded to test the level of transcription of DLX1 by reverse transcription quantitative PCR (RT-qPCR). Expression of the DLX1 sense transcript did not correlate with DLX1 methylation and did not significantly differ in forebrains between patients and controls (Fig. 4a).

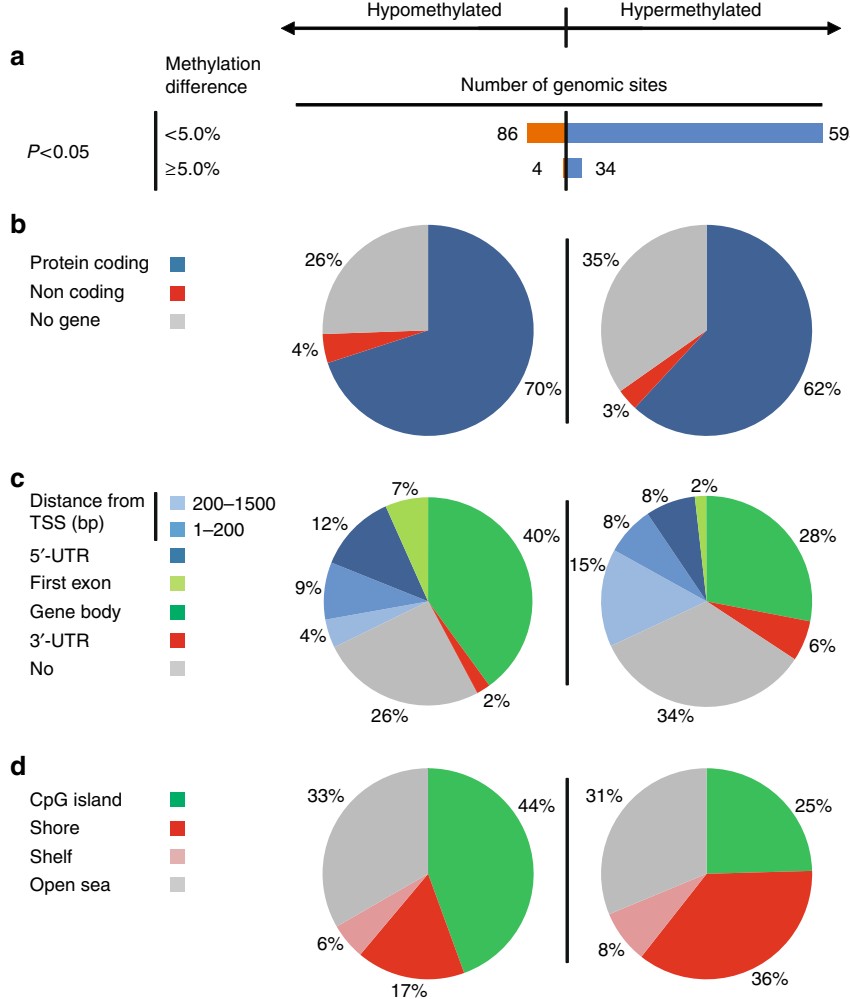

**Fig. 1** Epigenome-wide methylation analysis. **a–d** CpG sites hypo- and hypermethylated in $n = 94$ PSP patients vs. $n = 71$ controls are displayed on the left and right panels, respectively. **a** Number of genomic sites aberrantly methylated at a difference of < 5% and ≥ 5% between patients and controls (Benjamini-Hochberg corrected $P < 0.05$). **b** Location of aberrantly methylated sites within protein-coding genes, non-coding genes and intergenic regions. **c** Location of aberrantly methylated CpG sites in relation to defined regions of genes (TSS: transcription start site; 5′ and 3′ UTR: 5′ and 3′ untranslated region). **d** Association of aberrantly methylated sites with CpG islands (shores are < 2 kb and shelves are 2–4 kb from a CpG island)

Several reports indicate the existence of a *Dlx1* antisense transcript (*Dlx1as*) in the mouse[17–19]. We found such a *DLX1* antisense (*DLX1AS*) transcript in cDNA from human brain. Based on the mouse sequence we predicted a homologous sequence in human DNA in silico. We used primer walking from the putative transcription start site 3′ of exon 3 of the sense transcript to different predicted *DLX1AS*-exons. We found several alternatively spliced transcripts of *DLX1AS* by sequencing different PCR products and were able to extend longest transcripts beyond exon 3 of *DLX1* (Fig. 3b and Supplementary Fig. 3). The hypermethylated CpG sites are located in the region of exon 3 of the *DLX1AS* gene. A recently described enhancer region of *DLX1* overlaps with exon1 of *DLX1AS*[19,20]. For detection of the antisense transcript a region encoded by *DLX1AS* exon1 was used since this exon is part of all splice variants of the gene (Supplementary Fig. 3).

Transcription of *DLX1AS* was significantly reduced in patients ($P < 0.001$) and expression values inversely correlated with the degree of methylation ($P < 0.001$, Fig. 4b). Delta Ct values, calculated according to Pfaffl et al.[21], shown in Fig. 4b correspond to a 0.64-fold expression of *DLX1AS* in PSP as compared to controls.

**Single-cell analysis in healthy human cortex.** *Dlx1* and *Dlx1as* are almost exclusively expressed in neuronal cells in the mouse[22] (Supplementary Fig. 4a). Using published RNA-sequencing data in single cells from healthy human cortex[23] we quantified the reads of *DLX1* and *DLX1AS* transcripts. Consistent with the mouse data, we found *DLX1* and *DLX1AS* expression mainly in neurons (Supplementary Fig. 4b, c). *DLX1* was expressed in 26.15%, *DLX1AS* in 15.39% and both were expressed in 2.31% of neurons (Supplementary Fig. 4d).

**DLX1 protein.** Expression of *DLX1* was analyzed at the protein level in total protein extracts of frontal lobes from PSP patients and controls.

Western blot analysis of eight PSP and eight control forebrains demonstrated that DLX1 protein levels do not differ between patients and controls in white matter (Fig. 4c and Supplementary Fig. 5). In gray matter, however, higher levels of DLX1 protein were detected in PSP as compared to controls (Fig. 4d and Supplementary Fig. 6).

The findings were confirmed by quantitative immunohistochemistry on histological sections of frontal lobes from an independent set of patients. Densitometry of DLX1-

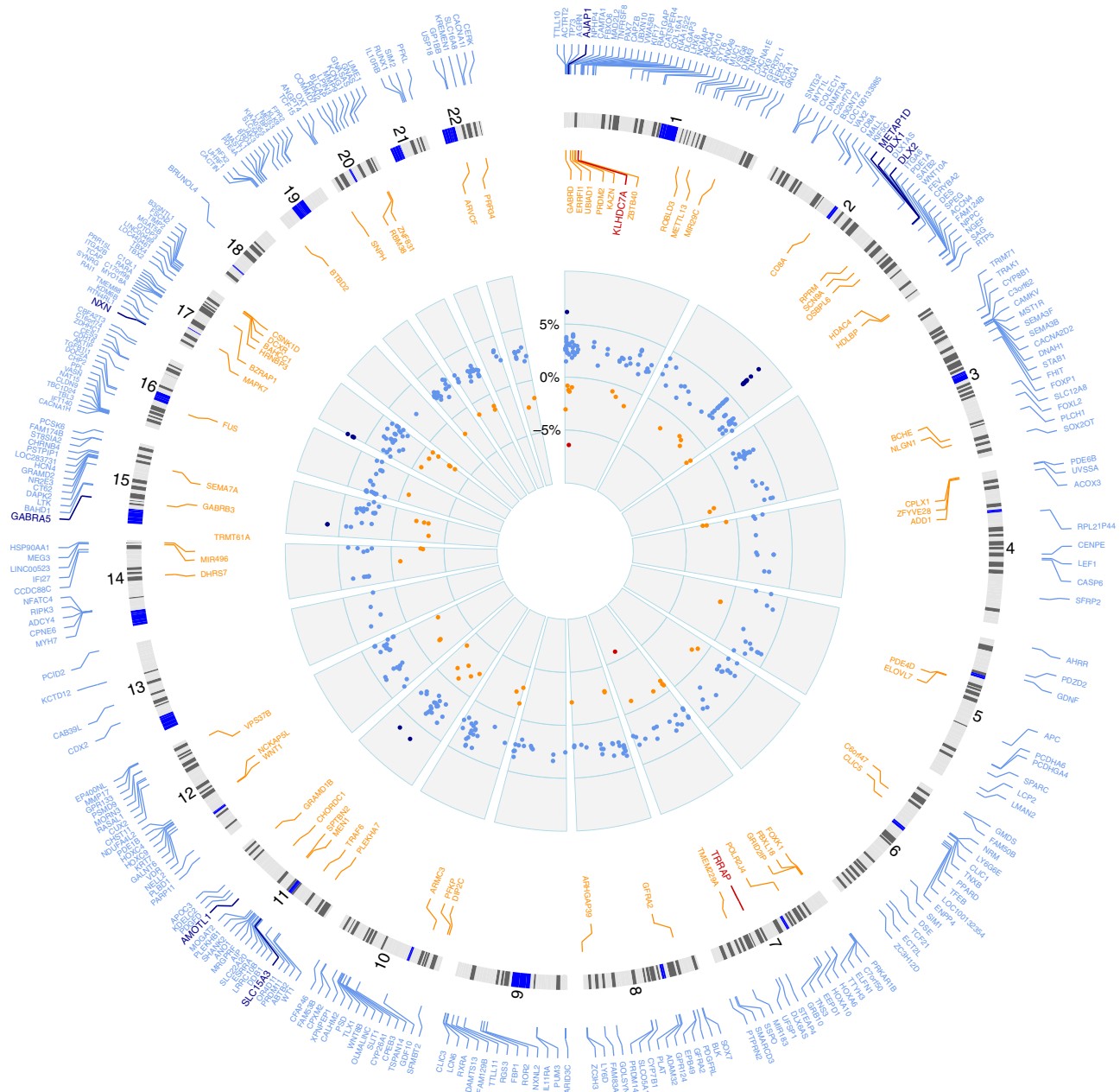

**Fig. 2** Circle plot of aberrantly methylated genes. $n = 451$ CpGs in protein-coding genes and $n = 26$ CpGs in non-coding RNA genes were found differentially methylated. After removal of duplicates, i.e., genes with more than one differentially methylated CpG, $n = 375$ genes show differential methylation in PSP patients. The two outer circles list the autosomal positions of $n = 375$ differentially methylated genes (Benjamini-Hochberg corrected $P < 0.05$; PSP vs. controls; hypomethylated < 5% (orange), ≥ 5% (red); hypermethylated < 5% (light blue), ≥ 5% (dark blue)). The intermediate circle depicts ideograms of the human autosomes head to tail. The inner circle displays the difference between hyper- (blue dots) and hypomethylation (orange/red dots) in percent at CpG sites in patients vs. controls. The highest degree of hypermethylation was detected at *DLX1/DLX1AS* (chromosome 2). This plot was generated according to Hu et al.[63]

immunoreactivity did not differ between patients and controls in white matter (Fig. 4e and Supplementary Fig. 7), but was significantly increased in gray matter of patients (Fig. 4f and Supplementary Fig. 8).

**Overexpression of *DLX1* and *DLX1AS* in Ntera2 and SH-EP cells.** In order to study the function of *DLX1* and *DLX1AS*, we transfected Ntera2 and SH-EP cells using eukaryotic expression vector (pcDNA-3.1-TOPO) containing either *DLX1* or *DLX1AS*. Initial experiments had shown that untreated Ntera2 cells express

less *DLX1* than SH-EP cells. *DLX1* was overexpressed 3–4-fold in Ntera2 cells (Fig. 5a) and *DLX1AS* was overexpressed 100–120-fold in SH-EP cells (Fig. 5b). We then proceeded to test the expression of known target genes of *DLX1*, i.e., *GAD1*[24], *GAD2*[24], *BRN3B*[25], *GnRH*[26], *and OLIG2*[27]. Of these genes *GAD1*, *BRN3B*, and *OLIG2* were upregulated in cells overexpressing *DLX1* (Fig. 5a) and downregulated in cells overexpressing *DLX1AS* (Fig. 5b). We also tested expression of *MAPT* that had previously been shown to play an important role in the development of PSP[4,5]. *MAPT* expression was reduced in cells overexpressing *DLX1* (Fig. 5a) and increased in cells overexpressing *DLX1AS* (Fig. 5b).

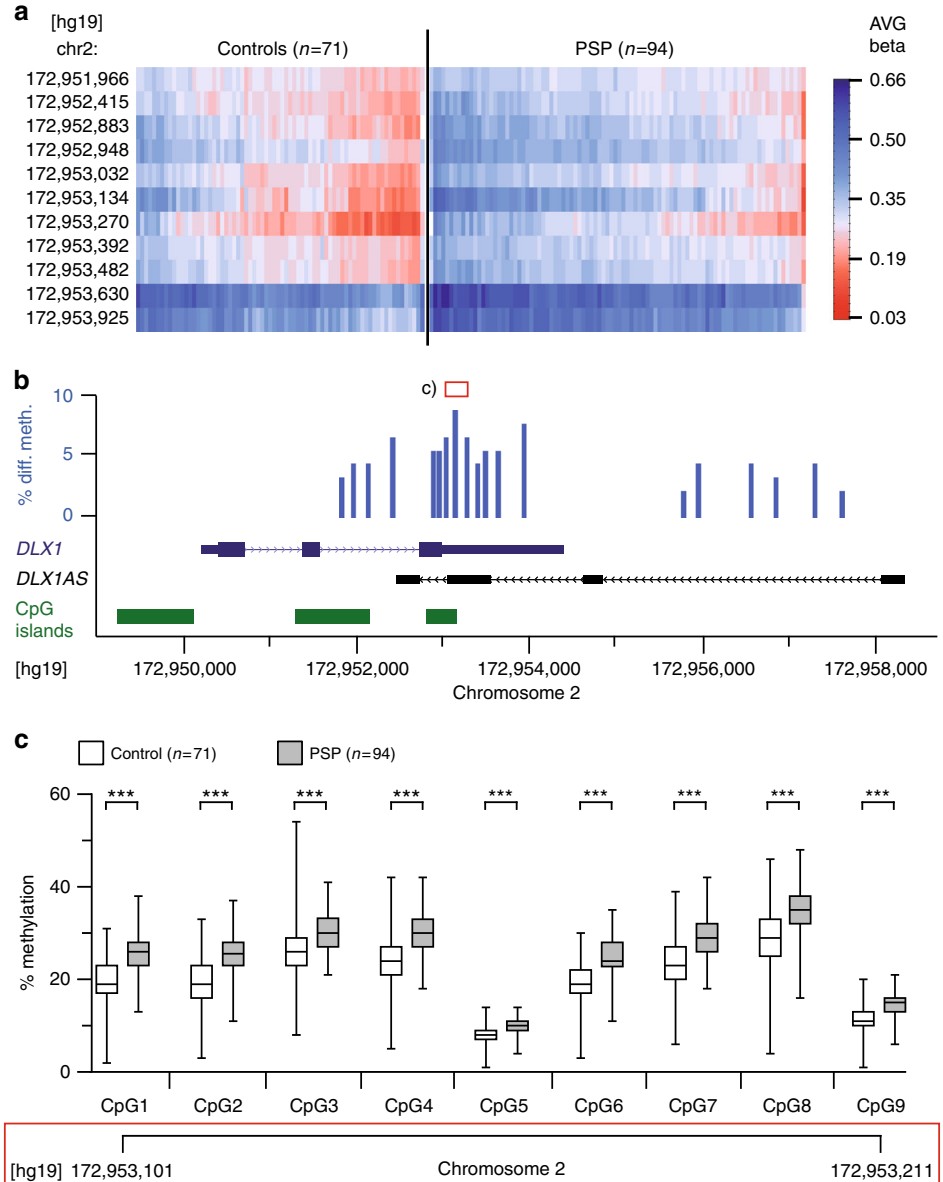

**Fig. 3** Methylation status of *DLX1*. **a** Heat-map showing degree of individual methylation at various genomic sites within *DLX1* on chromosome 2 (chr2) in forebrains of $n = 94$ PSP patients vs. $n = 71$ controls. Average (AVG) beta indicates the color-coded methylation value (1.00 equals 100%, Genome Studio Software Version 2011.1, Illumina, San Diego, CA). **b** *DLX1* is composed of three alternatively spliced exons (dark blue). *DLX1* antisense transcript (*DLX1AS*) is encoded by at least four alternatively spliced exons (black). The location of CpG islands relative to *DLX1* and *DLX1AS* is shown according to the UCSC genome browser data (green). The percentage difference in methylation in PSP as compared to controls at various sites within *DLX1* and *DLX1AS* is shown as bar chart (blue). **c** Pyrosequencing confirmed the differential methylation at nine CpGs within the CpG island of the 3′UTR of *DLX1* [red boxes in **b** and **c** indicate corresponding genomic regions; *** $P < 0.001$, Welch´s corrected unpaired *t*-test]. The line in the middle of the box and whisker graph represents the median (50th percentile). The box extends from the 25th to 75th percentile. The whiskers extend from the lowest to the highest value

**Knock-down of *DLX1* and *DLX1AS* in human striatal NPC**. In order to test whether a putative down-stream effect of *DLX1/DLX1AS* on *MAPT* also occurs in neural precursor cells (NPC) derived from human fetal striatum (strNPC) we transfected these cells with siRNAs that target *DLX1* and *DLX1AS*. As shown in Fig. 5c knock-down of *DLX1* resulted in significant upregulation of *MAPT* and of *DLX1AS*. Conversely, knock-down of *DLX1AS* resulted in downregulation of *MAPT* and in upregulation of *DLX1*.

We proceeded to test whether DLX1 affects Tau-dependent viability of strNPCs using the ATP firefly luciferase assay[28]. strNPCs overexpressing either 3R- or 4R-Tau were co-transfected with siRNAs directed against either *DLX1* or *DLX1AS*. As shown

in Fig. 5d, siRNA-mediated knock-down of *DLX1* decreased cellular viability, in particular of cells overexpressing 4R-Tau. Conversely, knock-down of *DLX1AS* significantly increased survival of strNPCs (Fig. 5d).

**Functional analysis of differentially methylated genes**. We performed in silico functional analyses of 375 different annotated genes that are represented by 451 (out of a total of 717) differentially methylated CpG sites on the Illumina 450 kb chip (Supplementary Data 2). Applying the Pathway Studio software and Fisher´s exact test we searched for enrichment of the Gene Ontology (GO) category "biological process" (Supplementary Data 4). *P*-values were corrected for multiple testing according to

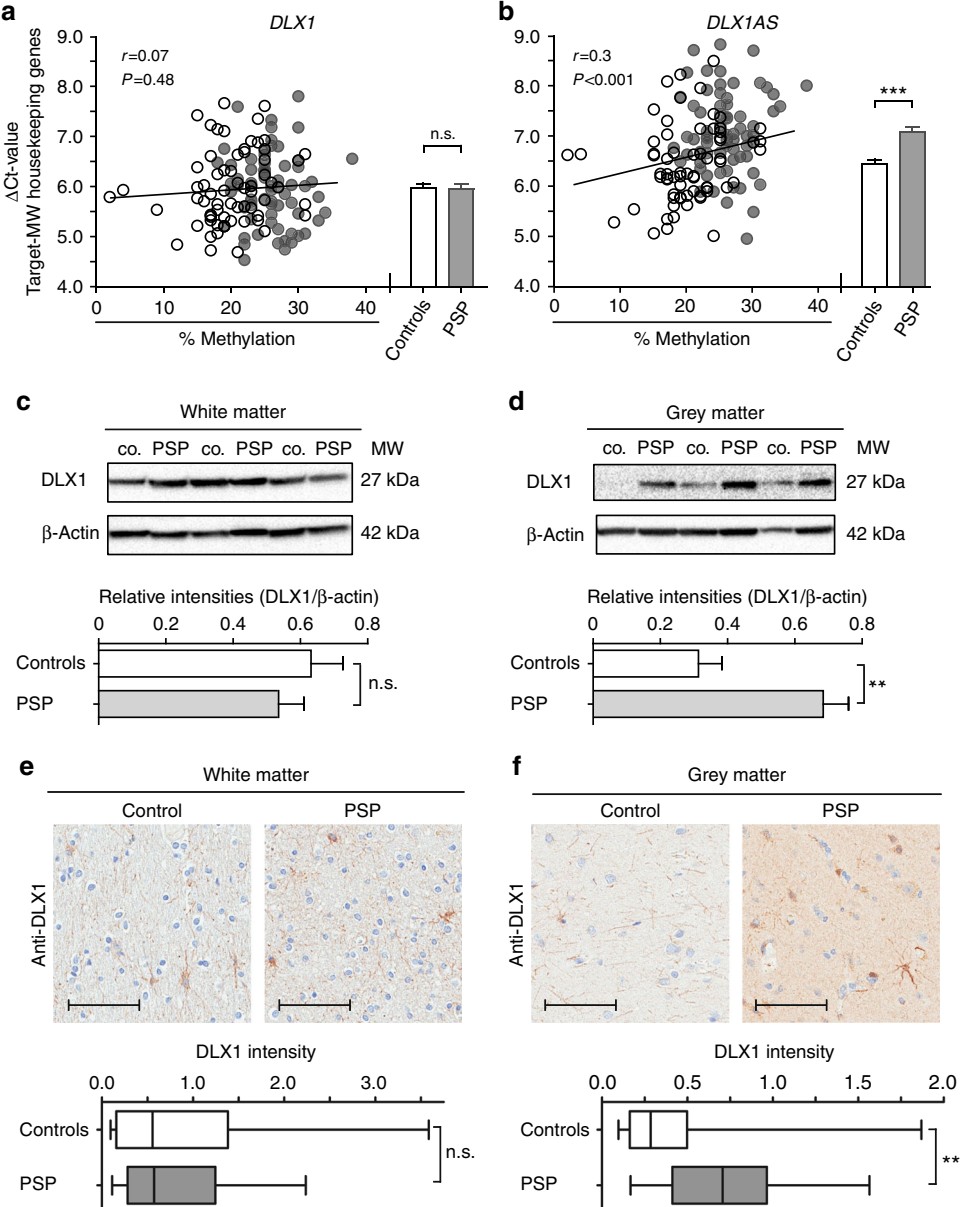

**Fig. 4** *DLX1* expression. **a** No correlation between expression of *DLX1* and degree of methylation in human forebrains (pyrosequencing value at CpG [hg19] chr2:172,953,097) [Pearson's correlation analysis including both PSP patients (*n* = 69, gray dots) and controls (*n* = 67, white dots)]. Expression of *DLX1* did not differ between patients and controls (Welch´s corrected unpaired *t*-test, n.s. = not significant, bar plot with mean and SEM). **b** Significant correlation between expression of *DLX1AS* and the degree of methylation. Expression of *DLX1AS* is significantly reduced in patients as compared to controls (***P < 0.001, Welch´s corrected unpaired *t*-test, bar plot with mean and SEM). **c**, **d** No difference between the amount of DLX1 protein in white matter of frontal lobe of PSP patients and controls (co.). **c** Significant increase of DLX1 protein in frontal lobe gray matter of PSP as compared to controls. **d** (*n* = 8 per group, **P < 0.01, Welch´s corrected unpaired *t*-test, bar plot with mean and SEM). β-Actin was used as loading control. **e** No difference in immunoreactivity of DLX1 in white matter of gyrus frontalis between PSP and controls. **f** Significant increase of DLX1 protein in frontal lobe gray matter of PSP patients as compared to controls (*n* = 24 PSP, *n* = 9 controls, **P < 0.01, Mann–Whitney Test). Scale bar: 100 μm. The line in the middle of the box and whisker graph represents the median (50th percentile). The box extends from the 25th to 75th percentile. The whiskers extend from the lowest to the highest value

Benjamini and Hochberg[14]. The top 20 highly significantly enriched categories include important functions pertinent to *DLX1* and *DLX2*, i.e., "anatomical structure development" (GO ID 48856), "regulation of signaling" (GO ID 23051), "cell fate commitment" (GO ID 45165), "positive regulation of transcription" (GO ID 45893) and the "cell surface receptor signaling pathway" (GO ID 7166). All 18 genes assigned to the term "Wnt signaling pathway" (GO ID 16055), which show a significantly

corrected *P*-value of $6.57 \times 10^{-04}$ are part of the "cell surface receptor signaling pathway" (Supplementary Data 4). Particularly many neuronal functions and pathways are distributed across the list of significantly enriched categories, i.e., "neuron fate commitment" (GO ID 48663), "cerebral cortex GABAergic interneuron fate commitment" (GO ID 28193), and "negative regulation of neurogenesis" (GO ID 50768) (all of which include *DLX1* and *DLX2*, Supplementary Data 4).

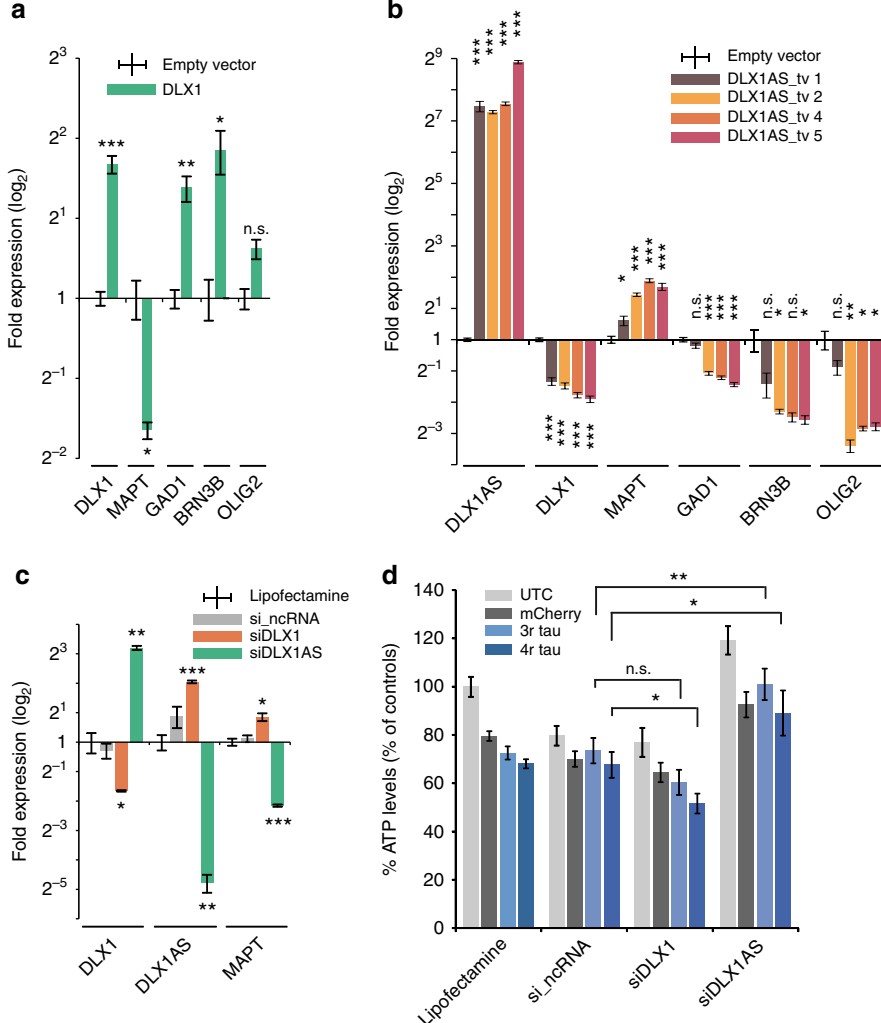

**Fig. 5** Overexpression and siRNA-mediated knock-down of *DLX1* and *DLX1AS*. **a** Overexpression of *DLX1* in Ntera2 cells results in upregulation of the *DLX1*-target genes *GAT1*, *BRN3B*, and *OLIG2* and in downregulation of *MAPT* (Student´s *t*-test). **b** Overexpression of *DLX1AS* in SH-EP cells using four different transcript variants (tv) (see Methods and Supplementary Fig. 3) results in downregulation of *DLX1* and its target genes and in upregulation of *MAPT* (Student´s *t*-test). **c** Specific siRNA-mediated knock-down of *DLX1* (siDLX1) in human fetal striatal neuronal precursor cells (strNPCs) results in upregulation of *MAPT* as compared to non-specific siRNAs (si_ncRNA). Conversely, knock-down of *DLX1AS* (siDLX1AS) causes upregulation of *DLX1* and downregulation of *MAPT* (Student´s *t*-test). **d** ATP assay showing decreased viability of strNPCs overexpressing 4R-Tau protein after siRNA-mediated knock-down of *DLX1*. *DLX1* knock-down does not significantly reduce viability of un-transfected cells (UTC) or of cells overexpressing either 3R-Tau or the control protein mCherry. Knock-down of *DXL1AS* increases viability in cells overexpressing 3R- or 4R-Tau (two-tailed, unpaired Student´s *t*-test, *P*-values: *$P < 0.05$, ** $P < 0.01$, *** $P < 0.001$, n.s. = not significant)

**Pathway analysis of differentially methylated genes**. Pathway analysis revealed interdependence of several genes found to be differentially methylated in PSP (Fig. 6 and Supplementary Data 5). Based on literature mining we propose two main possible pathways linking *DLX1* and *MAPT*:

(1) Activation of *MAPT*-encoded Tau protein via the Wnt signaling pathway: This notion is supported by the finding that DLX2 bound to Necdin activates the *WNT1* promoter[29]. In PSP patients, *DLX1* and *DLX2* are highly ($\geq 5\%$) and the WNT ligand family members *WNT10A*, *WNT8b* are distinctly ($> 3\%$) hypermethylated. Several additional differentially methylated genes (methylation differences $> 1\%$) are members of the WNT signaling pathway as well (Supplementary Data 4). Furthermore, WNT signaling appears to affect Tau phosphorylation in Alzheimer's disease[30–32].

(2) Tau phosphorylation via GABA(A) receptors: DLX1, DLX2, and GABA(A) receptors (encoded by the differentially methylated genes *GABRA5*, *GABRB3*, and *GABRD*) are members of the GABAergic interneuron-related network in humans[33]. Within this network DLX1/DLX2 regulate GABA synthesis[24]. Expression changes of *DLX1/DLX2* can alter activation of GABA(A). GABA(A) receptors in turn play an important role in Tau phosphorylation[34]. This observation is consistent with the notion that Tau may be affected via DLX1/DLX2 – GABA(A) in PSP.

Interestingly, we also detected two specific DLX1-binding sites in the *MAPT* promoter. One is located 1972-bp upstream of the TSS. The previously not described DLX1-binding motif (CAT-AATTAAAAT) was detected using the DiAlign TF program (Genomatix)[35] by utilizing an optimized matrix similarity. It was found in the *MAPT* promoter of human and rhesus monkey. At a

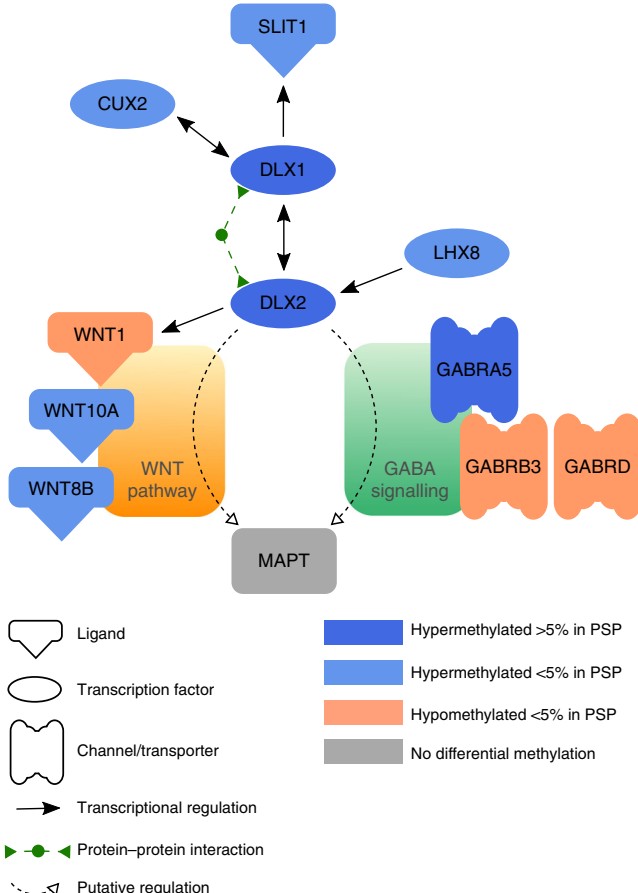

**Fig. 6** Pathway analysis. The pathway proposed was deduced from in silico literature mining for functional interactions of the differentially methylated genes. The network was consolidated by verification of each interaction in the published literature. Hypermethylated genes are depicted in blue and hypomethylated genes are given in orange. Note that DLX1/DLX2 may influence *MAPT* either via the WNT (brown) or via the GABA signaling (green) pathway (for details see text)

slightly lowered threshold of the optimized matrix similarity of 2% a second DLX1-binding site (TCTAATTTAAGA) was identified 550 bp upstream of the human TSS of *MAPT*. It is evolutionarily more conserved than the site 1972-bp upstream of the TSS. Apart from primates (humans and rhesus monkey) it is also found in other mammalian species such as cow and horse (Supplementary Fig. 9).

## Discussion

This epigenome-wide association study in brains of PSP patients and controls interrogated > 485,000 CpG sites representing 99% of RefSeq genes. Significant differential DNA methylation between patients and controls was found at 717 CpG sites. Four-hundred fifty-one of these sites correspond to 375 annotated genes. Most methylation differences in PSP were subtle (< 2% at $P < 0.05$), but greater differences ($\geq 5\%$) were observed at 38 sites representing ten genes. The degree of differential methylation in PSP was smaller than in cancer, where large differences in methylation are typically found[36] but comparable to other complex disorders such as multiple sclerosis[37,38] or Alzheimer's disease[39–41].

Genes involved in neuronal development and function were significantly overrepresented among the genes differentially methylated in PSP (Supplementary Data 4 and Supplementary

Data 5). These changes might have cumulative effects on disease origin and progression.

Two prior studies performed targeted, hypothesis-driven epigenetic analyses in DNA extracted from brain tissue and found differential methylation of *MAPT* on chromosome 17q21.31 in PSP patients[7,8]. Another prior study found differential methylation in the region of *MAPT* in peripheral blood DNA of PSP patients[9]. This methylation difference was associated with the H1 haplotype of *MAPT*, which is overrepresented in PSP[1,5]. In our epigenome-wide analysis, nine CpG sites within *MAPT* yielded significantly different methylation values in PSP vs. controls, which, however, did not hold up to correction for multiple testing (Supplementary Data 3). Thus, our findings are not at odds with these prior reports, but represent a more conservative interpretation.

Among the 375 genes identified in our study, differential methylation was most pronounced at *DLX1* that was hypermethylated at multiple CpG sites including a CpG island. Its methylation status differed by up to > 10% between PSP and controls. The other genes showed altered methylation levels at one or a few CpG sites only, and methylation differences were mostly low (< 2%). The observed hypermethylation of *DLX1* in its 3′region did not alter *DLX1* transcription. However, the amount of the newly detected *DLX1AS* transcript that overlaps with parts of exon 3 and with the 3′UTR of *DLX1*, was significantly reduced to 0.64-fold in PSP as compared to controls. *DLX1AS* is a likely negative regulator of *DLX1* translation by dimer formation between the sense and antisense RNA[42]. Consistently, we observed increased expression of DLX1 protein in gray matter of forebrains in PSP on western blots and by immunohistochemistry. Interestingly, an enhancer has been assigned to the 5′region of *DLX1AS*[18]. This enhancer appears to be part of *DLX1AS*. As such it might be involved in modification of the amount of DLX1 protein in different tissues (e.g., gray vs. white matter) by controlling the amount of *DLX1/DLX1AS* RNA dimer formation. Alternatively, instead of forming a RNA/RNA dimer, *DLX1AS* might also form a complex with various proteins that in turn regulate expression of *DLX1*. This mode of action has been shown for *DLX6AS/EVF2*[43,44].

*DLX1* encodes a homeobox-containing transcription factor. It is mainly expressed in GABAergic inhibitory interneurons. In concert with other *DLX* genes it impacts interneuron development and function[45–47]. This is consistent with our observation of mainly neuronal expression of *DLX1* in single cells in humans, confirming previously described neuronal specificity of *Dlx1* expression in mice[22]. Similarly, *Dlx1as* and *DLX1AS* are also predominantly expressed in neurons in both mouse and human. Interestingly, individual neurons of healthy human cerebral cortex expressed either *DLX1* or *DLX1AS*, but only very few neurons expressed both *DLX1* and *DLX1AS*, suggesting that expression of *DLX1* and *DLX1AS* might be regulated in an opposite manner (Supplementary Fig 4). This interpretation would be consistent with our cell culture experiments showing increased *DLX1AS* expression upon *DLX1* silencing and vice versa.

The finding of differential methylation of various genes within a putative *DLX1* pathway (Fig. 6) further strengthens the notion of an important role of *DLX1* in the etiology of PSP. While methylation differences were comparatively subtle at most of these genes, together they might be sufficient to alter the pathway in a functionally relevant manner in PSP. During normal ageing, the methylation pattern of cells may change dramatically, without having apparent functional effects in most cases[48,49]. In some instances, however, functionally related genes may become differentially methylated and predispose to disease via their interaction. Of particular interest is our finding that the putative *DLX1* pathway extends to *MAPT*. Via this pathway overexpressed DLX1

might modify protein Tau by increasing its phosphorylation, which in turn contributes to tangle formation[1]. In vitro experiments support dependency of *MAPT* expression on the concentration of *DLX1/DLX1AS* transcripts. Currently, it remains unclear whether *DLX1* influences *MAPT* expression directly or indirectly. Given that we could not conclusively show direct binding of DLX1 to the *MAPT* promoter, the effect of *DLX1* on *MAPT* appears to be more indirect (via the Wnt or the GABAergic interneuron-related network pathway). Furthermore, a high-throughput ChIP-sequencing study with enrichment profiles for a large set of factors[50] also used an antibody against DLX1 but did not find enrichment at or near the *MAPT* locus. Differential methylation of the various genes related to neuronal development and function might also be a secondary event rather than the primary cause of disease in PSP. It has been shown, that DLX1 is essential for survival of adult interneurons in the neocortex[46] and in mature retinal cells[51]. Thus, differential regulation of DLX1 and interacting genes involved in neuronal development, differentiation and plasticity, might refer to endogenous repair mechanisms in neocortical areas affected by PSP[52]. Consistent with these findings we found that siRNA-mediated knock-down of DLX1 decreases neuronal viability in strNPCs overexpressing either 3R-Tau or 4R-Tau. Knock-down of *DLX1AS* rescued cells from death independent of overexpression of Tau.

In conclusion, we found significant methylation differences in DNA from forebrains of PSP patients as compared to controls. Differential methylation affected a high percentage of genes involved in neuronal differentiation and function. Methylation differences were particularly high at *DLX1/DLX1AS* that are expressed in neurons almost exclusively. DLX1 protein was increased in gray matter of patients´ forebrains and was shown to negatively affect *MAPT* expression in vitro. DLX1 may thus contribute to disease by disturbing the normal function of neurons and might serve as a novel molecular target for the development of disease-modifying therapies.

## Methods

**Brain tissue.** The use of human brain tissue for this project was approved by the ethics committees of the University of Giessen and of the Technical University of Munich. Brain samples were from patients who had given informed consent before death.

For the epigenome-wide DNA methylation analysis, postmortem prefrontal lobe tissue of PSP patients ($N = 94$, $72 \pm 5.3$ years of age, $N = 54$ (57%) male) and controls without neurological or psychiatric diseases ($N = 71$, $76 \pm 7.9$ years of age, $N = 46$ (67%) male) was obtained from the CurePSP brain bank, Mayo Clinic, Jacksonville, Florida and the Victorian Brain Bank, Carlton, Australia. Detailed information on clinical and neuropathological findings is given in Supplementary Data 1. Of the PSP patients, 83 were homozygous and 11 were heterozygous for allele H1 of *MAPT*, none were homozygous for the H2 allele. Among the controls, 54 were homozygous for the H1 allele, 14 were heterozygous and 3 were homozygous for the H2 allele.

For western blot analysis, postmortem gyrus frontalis superior tissue of PSP patients ($N = 8$, $71.9 \pm 6.3$ years of age, $N = 5$ (63%) male) and controls without neurological or psychiatric diseases ($N = 8$, $74.6 \pm 7.0$ years of age, $N = 2$ (25%) male) was obtained from the Ludwig-Maximilians-Universität Munich and the Hospital Clínic de Barcelona brain banks.

For immunohistochemistry, postmortem gyrus frontalis superior tissue of PSP patients ($N = 24$, $72.6 \pm 4.8$ years of age, $N = 18$ (75%) male) and controls without neurological or psychiatric diseases ($N = 9$, $70.6 \pm 9.7$ years of age, $N = 4$ (44%) male) was obtained from the CurePSP brain bank, Mayo Clinic, Jacksonville, Florida.

**Nucleic acids and PCR.** DNA and RNA were extracted from tissue samples by standard procedures. DNA was prepared using the tissue extraction kit from Qiagen (Hilden, Germany) and RNA was extracted using the RNeasy Lipid Tissue-Kit (Qiagen). Genomic DNA was bisulfite-converted using the EpiTect Bisulfite Kit (Qiagen). For RT-qPCR, total RNA was converted into cDNA using the QuantiTec-Reverse-Transcription kit (Qiagen).

For investigation of the brain tissue samples, all primers used for standard PCR and RT-qPCR were designed using OLIGO Primer analysis software (Vers. 6.41; Molecular Biology Insights, Inc., Colorado Springs, CO, USA). Primers used for qPCR are given in Supplementary Table 1. Fluorescence data were compiled using

the CFX384 qPCR system from Bio-Rad (Hercules, CA, USA). Housekeeping genes *EIF4A2* and *CYC1* were co-analyzed for normalization of RNA content in each brain sample[53].

Quantification of expression differences (fold-expression) was calculated using the REST-Software-Package-2009 as described elsewhere[54].

Expression of the housekeeping genes *B2M*, *EIF4A2*, and *CYC1* was analyzed for normalization in cell culture experiments with SH-EP und Ntera2 cells. Stability values, i.e., coefficient of variation (CV-value) and gene stability value (*M*-value) for each housekeeping gene were < 0.25 for CV- and < 0.5 for the *M*-value. Fold expression was calculated with the integrated qBase module of the Bio-Rad CFX Manager 3.1 (3.1.1517.0823) software. Significance of expression changes was evaluated by unpaired two-tailed Student´s *t*-test.

We have used DNAse treated total-RNA in all RT-qPCR experiments. This excluded co-amplification of DNA that might have contaminated RNA preparations. Exclusion of DNA co-amplification was particularly important for analysis of exon1-derived RNA of *DLX1AS*. Exon1 RNA of *DLX1AS* was analyzed since this exon was present in all alternative transcripts.

RT-qPCR analyses were performed on RNAs extracted from strNPCs using SYBR Green Select qPCR Supermix (#4472954, Life Technologies, Carlsbad, USA), 5 ng cDNA synthesized from total RNA, 0.2 µM forward and reverse primers (primer sequences are given in Supplementary Table 1). qPCR analysis was performed on a Step One Plus instrument (Thermofisher Scientific, Carlsbad, USA). Initial incubation at 50 °C for 2 min and an additional 2 min at 95 °C was followed by 40 cycles of 15 s at 95 °C and 60 s at 60 °C. Threshold cycle (CT) values were set within the exponential phase of the PCR. Data were normalized to five housekeeping genes: *TBP*, *GPBP1*, *PPIA*, *PSMC1*, *UBQLN2* and the comparative normalized relative quantity (CNRQ) was used to calculate fold expression (qBase®, Biogazelle, Belgium). Gene regulation was statistically evaluated by two-tailed Student's *t*-test on the assumption of equal variances.

**Array analysis.** The Infinium 450k array of Illumina Inc. (San Diego, CA) was used for methylation analysis. This array analyzes more than 485,000 CpG sites distributed over the entire genome. Two-hundred nanograms of bisulfite-converted DNA were hybridized to the arrays according to the manufacturer´s instructions. In order to exclude batch effects, we alternated hybridization of PSP and control samples. Arrays were scanned on an Illumina iScan platform at the Life & Brain Center (Bonn/Germany). Array data were evaluated, preprocessed and normalized using the ChAMP pipeline (V.2.8.1)[55,56] with minor modifications. Specifically, raw array data were uploaded to the ChAMP pipeline using the minfi option[57]. Relative proportions of neuronal and non-neuronal cells in each sample were estimated based on the raw data applying the compositeCellType = "DLPFC" option of estimateCellCounts function of the R Bioconductor minfi package[58]. Probes with less than three measured beads or a detection *P*-value > 0.01 as well as probes interrogating CpGs that fall on or near to a SNP were removed based on recommendations by Zhou et al.[59]. Similarly, probes aligning to multiple locations as defined by Nordlund et al.[60] were removed from the analysis. All samples had more than 99.3% of valid probes, therefore, no samples were removed from the analysis based on the quality of the data. The sex of each sample was verified using an in-house R script based on the DNA methylation profile of the X and Y chromosomes. One single sample was not in concordance with the assigned sex and was excluded from further analysis. Differences between probes due to InfI and InfII probe usage as well as between samples were normalized using BMIQ[61] on the 435,803 remaining probes. Differential methylation was determined using a modified version of the champ.dmp function with age, sex, and estimated proportion of non-neuronal cells as covariates in a linear regression analysis based on the limma package[62] adjusting *P*-values for multiple testing with a Benjamini-Hochberg correction[14]. Differentially methylated CpGs were considered significant at adjusted *P*-values < 0.05. To account for potential confounding owing to genetic variation, significant probes were filtered for significant mQTLs in adult prefrontal cortex using previously published data, which removed three CpGs all of which associated with *GABRA5* (cg01378667, cg03325535, cg10318222) as defined by a SNP in close proximity to the gene (rs7496866)[15].

We computed the exact statistical power for each of the 485,577 probe sets on the microarray based on the data obtained in our sample by applying the function pwr.t2n.test of the pwr package of R statistical software. The parameters of this function are the number of patients and controls, i.e., $n = 94$ and $n = 71$, respectively, as well as the significance level of 0.05 and Cohen's d (effect size). The effect size is defined by the difference between the means of the group divided by the pooled standard deviations of the two groups. Given these parameters, a power of at least 80% is calculated for 14,553 out of the total of 485,577 probes. Of the 717 CpG sites, which were significantly differentially methylated in PSP as compared to controls 664 (92.6%) showed a power of greater than 80% (Supplementary Data 6).

The circle plot was built with hg19 as reference genome using the OmicCircos R-Package Version 1.4.0[63] for R version 3.1.3[64].

**Pyrosequencing.** Pyrosequencing was done on a Pyromark-Q24 using PyroMark-Q24 tools and reagents (Qiagen) according to the manufacturer´s protocol. Primers for pyrosequencing (PCR product and sequencing primer) were designed using PyroMark-Assaydesign-2.0.1.15 (Qiagen) software. Primer sequences are shown in Supplementary Table 1. Data were analyzed with the PyroMark-Q24 software.

**Western blot**. White and gray matter from the Gyrus frontalis superior of post-mortem tissue was extracted in ice-cold N-PER™ lysate buffer (#87792, ThermoFisher Scientific, Waltham, MA, USA) using the Dounce homogenizer with 10–20 strokes on ice. Proteins were obtained after removal of cellular debris by centrifugation at 10.000 rpm for 10 min at 4 °C. Thereafter, 1 × HALT™ protease inhibitor (#78442, ThermoFisher Scientific, Waltham, MA, USA) was added to each supernatant. Protein concentration was determined by Bicinchoninacid assay (#23225, ThermoFisher Scientific, Waltham, MA, USA).

Equal amounts of proteins were separated by sodium dodecyl sulfate polyacrylamide gel electrophoresis with 4–20% Criterion™, TGX Stain-free™ Protein gels (#5678094, Bio-Rad, Munich, Germany). Molecular weight markers used were # 830537 and # 830552 (Hessisch Oldendorf, Germany). Thereafter the protein staining was activated by UV light using the ChemiDoc-XRS system (Bio-Rad, Munich, Germany). After transfer to immuno-blot polyvinylidene difluoride (PVDF) membrane (#162–0177, Bio-Rad, Munich, Germany) proteins were fixed on the membrane with 3.75% paraformaldehyde (PFA) in 1 × PBS at room temperature (RT) for 15 min. After 3 × 10 min washing steps in 1 × PBS, membranes were incubated with primary DLX1 antibody (dilution 1:1000, ab126054, polyclonal rabbit AB, Abcam, CA, UK) for 48 h at 4 °C. (Human Protein Atlas entry: [https://www.proteinatlas.org/ENSG00000144355-DLX1/tissue]). Positive bands were detected using HRP-conjugated secondary antibody (dilution 1:5000, ab P0448, polyclonal goat anti rabbit immunoglobulins, affinity isolated, Agilent, CA, USA) and chemolumina substrate (#170–5061, Clarity™ Western ECL Blotting Substrate, Bio-Rad, Munich, Germany). Blots were stripped and hybridized with β-actin antibody overnight (dilution 1:2000, ab 3700, clone 8H10D10, mouse monoclonal β-actin antibody, Cell Signaling Technology, CA, UK). Positive bands were detected by an HRP-conjugated secondary antibody (dilution 1:5000, ab P0447 polyclonal goat anti mouse immunoglobulins, affitinity isolated) and chemoluminescent substrate (see above). Bands were scanned using the ChemiDoc-XRS system and Image Lab software V.5.1 was applied to measurement of optical densities (Bio-Rad). Optical densities of anti-DLX1 antibody positive bands were compared to TGX labeled total protein and the housekeeping protein β-actin. DLX1 antibody was validated in positive and negative controls, e.g., tissue specific human neuronal progenitor cells derived from striatum or protein lysates from the human pancreas.

**Aperio methodology**. DLX1 immunopositive structures on tissue sections were quantitatively measured applying Aperio technology (Leica Microsystems Inc., Buffalo Grove, IL). Immunostained frontal cortical sections were scanned at 20 × magnification on the ScanScope® AT2 (Leica Biosystems, Wetzlar, Germany). The tissue sections were annotated using ImageScope (Leica Biosystems) to trace both gray matter and neighboring white matter along the strait of the gyrus, specifically avoiding the depth of the sulcus and rise of the gyrus. Annotated slide files were analyzed with a custom-designed color deconvolution algorithm using the eSlideManager (Leica Biosystems).

**Cell culture experiments**. We used Ntera2, SH-EP cells, and strNPCs in our studies. Ntera2 cells (pluripotent human embryonal carcinoma cells with characteristics of neuroepithelial precursor cells) were applied to overexpression experiments of DLX1 since intrinsic DLX1 mRNA expression is low. Ntera2 cells display neuronal characteristics (e.g., differentiation into neurons upon induction with retinoic acid), grow well under standard cell culture conditions and can be easily transfected.

SH-EP cells, a human neuroblastoma-derived cell line, were used in experiments involving DLX1AS overexpression. Endogenous expression of DLX1 is increased as compared to other cell lines tested. We studied human striatal neuronal precursor cells (strNPCs) for siRNA experiments since they are commonly used as proxies of neuronal cells. They are not transformed and thus lack the malignant phenotype of other cell lines.

Human cell lines SH-EP were grown in RPMI Medium (GibCo) and Ntera-2 (NT2) in DMEM Medium (GibCo), containing 10% fetal calf serum (SIGMA), Glutamine [2 mM], Penicillin-G [100 U/ml] and Streptomycin [100 μg/ml] at 37 °C and 5% CO2. (SH-EP and NT2 Cells were provided by the Neuroblastoma Study Group Cologne, Germany).

A full-length cDNA encoding DLX1 (NM_178120.4) and cDNAs of DLX1AS transcript variants (tv) 2 (KU179669), 4 (KU179671), 5 (KU197672) and partial transcript variant 1 (5´part of KU179668 [hg19]chr2:172,958,068–172,958,335 joined with chr2:172,954,669–172,954,850) were PCR-amplified from fetal brain mRNA, sequenced and cloned into a pcDNA-3.1-TOPO vector.

Cells were transfected with these constructs or empty vector using Viromer-Red transfection reagent (Lipocalyx, Germany) and selected in medium containing G418. Cells were grown in this medium for 14–21 days. They were harvested at a confluency of 70–80%.

siRNA-mediated knock-down experiments were performed in fetal striatum-derived human neuronal progenitor cells (srtNPCs)[65] using siPOOL-DLX1 siRNA, siPOOL-DLX1AS and siPOOL non-coding (nc)siRNA (siTOOLs BIOTECH, Planegg, Germany). Experiments were repeated three times. Transfection of siRNA-Pools was done with Lipofectamine® RNAiMAX reagent (#13778150, ThermoFisher Scientific, Carlsbad, USA) according to the manufacturer's protocol

at a final concentration of 5 nM. Cells were harvested and lysates prepared after 6 days of culture.

ATP Assay was performed according to standard procedures[27]. strNPCs were transfected with siPOOL-DLX1, siPOOL-DLX1AS or siPOOL-ntsiRNA on day one of culture. After 48 h Lentiviruses overexpressing either mCherry, MAPT3R or MAPT4R[66,67] were added to the cells. Cells were harvested 6 days after onset of culture and ATP assays were performed using ViaLight™ plus kit (LT07-221, Lonza, Germany).

**Literature mining and pathway analysis**. Using Network Builder of the Pathway Studio software (Elsevier) version 12.0.1.5 we generated a literature-derived network that was based on text mining for direct interactions between significantly differentially methylated input genes. The input dataset comprised genes adjacent to the 717 genomic positions (CpG sites) found to significantly differ in methylation between PSP patients and controls. We only analyzed those genes by Network Builder that are known to be expressed in the telencephalon and in interneurons. The GeneRanker program (Genomatix) was used for analysis of association of differentially methylated genes in tissues (Supplementary Data 5)[67]. The literature-derived network was curated and extended by manual literature searches as well as by in silico prediction of transcription factor binding sites applying the MatInspector program (Genomatix).

The complete set of differentially methylated genes was further explored using the Pathway Studio software for enrichment in the category "biological process" of the Gene Ontology by applying Fisher's exact test. P-values were corrected for multiple testing by Benjamini and Hochberg[14].

**Single-cell analysis**. We downloaded data on single cells from GEO and processed them according to the recommendations detailed in [GSE67835][22].

We applied the program Prinseq (-min_len 30) to remove very short non-specific reads. Prinseq also trimmed both ends of the reads in order to eliminate 5´duplicates (-trim_left 10) and to remove low quality 3´ends (-trim_qual_right 25). Furthermore Prinseq filtered reads of low complexity (-lc_method entropy /-lc_threshold 65). The program FASTQC was used to identify sequences that are overrepresented (adapter) in order to exclude them from further analysis. We used the Prinseq tool to remove orphan pairs less than 30 bp in length followed by removal of nextera adapters using Trim Galore (--stringency 1).

Reads were aligned to the hg19 genome with STAR using the following options (-outFilterType BySJout/--outFilterMultimapNmax 20/--alignSJoverhangMin 8/--alignSJDBoverhangMin 1 /--outFilterMismatchNmax 999/--outFilterMismatchNoverLmax 0.04/--alignIntronMin 20 /--alignIntronMax 1000000 /--alignMatesGapMax 1000000 /--outSAMstrandField intronMotif).

Aligned reads were converted to counts for each gene using HTSeq (-m intersection-nonempty /-s no). The human Ensembl General Feature Format (GTF) annotation file (version 2013–09) necessary for HTSeq was extended by DLX1AS splice variants represented by the following Genbank identifier KU179668.1, KU179669.1, KU179670.1, KU179671.1, and KU179672.1. For all analyses Genome_build hg19 was used. Finally, counts were converted to FPKM (Fragments per kilobase of transcript sequence per million mapped fragments) values.

Out of 466 available single-cell datasets, we selected 251 datasets that represent specific cortical cell types (astrocytes, microglia, oligodenrocytes, endothelia, neurons, oligodendrocyte precursor cells, see [https://www.ncbi.nlm.nih.gov/Traces/study/?acc = SRP057196]). Hybrid cells and fetal quiescent cells were not considered.

**In silico promoter analysis**. All sequences analyzed are from the promoter sequence retrieval database EIDorado 12–2016 (Genomatix) that is based on NCBI build GRCh38. The following Genomatix / Entrez Gene identifiers for MAPT were used: GXP_10577 / 4137 (human), GXP_7225047 / 574327 (rhesus_monkey), GXP_5397266 / 100054638 (horse), GXP_3861378 / 281296 (cow). Promoter sequences of MAPT from four mammalian species were aligned using the DiAlign TF program of the Genomatix software suite GEMS Launcher in order to evaluate overall promoter similarity and to identify DLX1-binding sites. The corresponding position weight matrices V$DLX1.01 and V$DLX1.02 were applied to promoter analysis according to the Matrix Family Library Version 10.0 (October 2016). Binding sites were considered conserved if promoter sequences could be aligned in the region of the DLX1-binding site using the DiAlign TF program.

**Statistics**. Statistical tests were performed within the R computing environment[64] [http://www.r-project.org]. In order to calculate Pearson's product moment correlation the R function cor.test was used by setting alternative = "two-sided" and method = "pearson". We applied the function t.test to two sample t-tests with Welchs's correction and the arguments paired = "FALSE", alternative = "two.sided" and var.equal = "FALSE". Mann–Whitney tests were performed by using the R function wilcox.test in conjunction with paired = "FALSE" and alternative = "two-sided".

Test of normality according to Kolmogorov–Smirnov was applied separately to CpG-control and CpG-PSP samples for one specific gene from pyrosequencing analysis. If one of both tests was significant the corresponding dataset was defined

as not normally distributed. If only one of all CpG samples for a specific gene are not normally distributed the unpaired *t*-test with Welch's correction was performed otherwise the Mann–Whitney test. In case of the densitometry of DLX1-immunoreactivity experiment the Mann–Whitney test was used because the corresponding datasets were not normal distributed according to Kolmogorov–Smirnov.

In general, $P < 0.05$ was considered significant. For bar plots the mean ± SEM (standard error of the mean) is given unless indicated otherwise.

Whenever multiple tests were computed e.g., for enrichment analyses of gene occurrences in pathways these *P*-values were corrected according to Benjamini-Hochberg or according to other appropriate methods, which are given in the corresponding subsections for array and pathway analyses.

**Data availability**. Normalized and raw BeadChipArray data have been deposited in NCBI- Gene Expression Omnibus (GEO) with the accession code GSE75704.

*DLX1AS* transcript variants have been uploaded to NCBI under accession numbers KU179668, KU179669, KU179670, KU179671, and KU179672.

All other data are available within the paper and its associated supplementary material or upon reasonable request from the corresponding authors.

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

## Acknowledgements

We gratefully acknowledge technical support of Claudia Röhrsheim, Michaela Weis, and Stefanie Teschler (RT-qPCR, pyrosequencing), Monica Castanedes-Casey (immunohistochemistry), and Drs. Melissa Murray and Amanda Serie (Aperio technology). We thank Drs. Eilis Hannon and Jonathan Mill (University of Exeter Medical School) for the mQTL data of the adult prefrontal cortex. We also thank Drs. Per Hoffmann and Stefan Herms for advice on data analysis. Dr. Hong Xu is acknowledged for his help with DLX1 siRNA and tau overexpression experiments. We are indebted to Dr. Ellen Gelpi and Dr. Laura Molina at the Neurological Tissue Bank of the Biobank-Hospital Clinic-IDIBAPS, Barcelona, for data and sample procurement. We thank Dr. Hagen Scherb (Institute of Computational Biology, Helmholtz Center Munich, Germany) for the support regarding statistical questions. This work was supported by a grant (#518-14)

from CurePSP (to G.U.H., and U.M.), a German Federal Ministry of Education and Research (BMBF)—and French National Research Agency (ANR)—funded project on "Epigenomics of Parkinson's disease" (O1KU1403A/ ANR-13-EPIG-0003-05 EpiPD to G.U.H., J.T. and U.M.), the BMBF-funded project "HitTau" (01EK1605A to G.U.H.), the Deutsche Forschungsgemeinschaft (DFG, HO2402/6-2 & Munich Cluster for Systems Neurology SyNergy) (to G.U.H.), the NOMIS foundation (FTLD project to G.U.H.), the German Science Foundation Collaborative Research Center (CRC) 870 and the Helmholtz Portfolio Theme "Supercomputing and Modeling for the Human Brain" (SMHB) (to W.W., D.T. and D.V.W.), and the Bayerisches Staatsministerium für Bildung und Kultus, Wissenschaft und Kunst within Bavarian Research Network "Human Induced Pluripotent Stem Cells" (ForIPS) (to G.U.H., S.C.S., W.W. and D.V.W.).

## Author contributions

A.W. planned the methylation study, performed sequencing and RNA expression experiments, analyzed the data and prepared the figures. S.C.S. performed the western blot and siRNA experiments. J.T. and F.B. performed the methylation data analysis. P.W. studied DLX1AS RNA transcripts. P.T. performed the immunohistochemistry experiments and J.S. established NPCs. D.T., D.V.-W., and W.W. contributed the gene ontology and pathway analysis. D.T. performed power analysis. A.C.W. prepared the circle plot. T.A., E.G., C.M., and J.v.S contributed brain tissue. D.W.D. contributed brain tissue and supervised the immunohistological experiments. T. Ad. performed chromatin immunoprecipitation. M.F. analyzed mQTL data. G.U.H. and U.M. conceived of and supervised the studies, coordinated assembly of the data and wrote the paper with input from all authors.

## Additional information

**Competing interests:** The authors declare no competing interests.

