## [Peer Review File · Nature Communications]

Reviewer #1 (Remarks to the Author):

The manuscript by Weber et al is a genome-wide DNA methylation study in post-mortem brain tissue of individuals with PSP and controls.

Clearly, the use of post-mortem brain tissue is a unique feature of this project and a major advantage in the study of neuropsychiatric disorders compared to the use of peripheral tissue. The manuscript is well written, concise and the finding of the DLX1 antisense transcript expression influenced by DNA methylation is intriguing.

However, there are several addressable concerns that limit my enthusiasm at this point:

Genotype: Although the authors test for known genetic variants in MAPT influencing the development of PSP, genome-wide genotyping and mQTL analysis should be performed. One first step could also be to check for SNPs that are known regulators of the CpG sites found in this study.

Power analysis: The overall sample size is small, although I recognize the value of the brain samples and the difficulties to obtain larger number of samples. However, with 94 vs 72 there is essentially no power to detect small differences unless the authors can show otherwise. I tried to calculate the exact power, however, some information are missing from the methods incl the number of CpG that were tested etc. The authors may want to consider a better data reduction strategy to reduce the number of test to correct for multiplicity.

Cell type specificity: The authors show desitometry and western blot data for DLX1. I was wondering if a deeper description of the DLX1 cells in the PSP tissue would be useful as an avenue for novel treatment approaches. Is the DLX1AS transcript expressed in all the DLX1 positive cells, what are other markers that define this population?

Functional impact: The authors present gene expression, western blot and other data related to DLX1. I was wondering of a causal proof of differential methylation at the DLX1/DLX1AS locus using CRISPR/Cas in cell culture could firmly establish the link between DNA methylation and expression differences on the mRNA and protein level.

Description of sample: The descriptions of the various post-mortem brain samples are insufficient (or basically not existing). There are no information on e.g. PMI, brain weight, medications, other diagnoses, hemisphere used, storage, RIN, ethnicity etc etc... I'm wondering if the authors even include the covariates in their methylations analyses? What covariates were included in the case-control design (not only for the methylation analyses but in general)? The methods did not mention gender and age, is this correct? Did the authors correct for batch effects (e.g. chip, bisulfite conversion?). A table with demographics is essential.

Reviewer #2 (Remarks to the Author):

Weber and colleagues profile 94 PSP and 72 control prefrontal samples for methylation changes using 450K BeadChips. 621 sites showed >1% while only 48 showed >5% change. Most changes were associated with protein-coding genes. Pyrosequencing confirmed some candidate sites. The authors found >5% hypermethylation at the 3'UTR region of DLX1 that overlapped with exon 3 of DLX1AS whose transcript was altered in the disease samples, correlating with the change in methylation, and is proposed to alter DLX1 translation and contribute to PSP. The number of samples is appropriate, as are statistical methods.

Little is known regarding PSP at a molecular level, especially any involvement of lncRNAs, The study therefore is useful for the field and may instigate more functional studies. I have a few suggestions to strengthen the manuscript:

1. To confirm qPCR results, as opposed to using qPCR primers only from DLX1AS exon 1 (as this enhancer region may be independently transcribed) authors could use primer pairs from the common sequence of DLX1 and DLX1AS as well as unique DLX1 3'UTR sequence. The qPCR results should be identical if these two regions are co-transcribed.

2. Direct binding of DLX1 to the predicted MAPT sites should be performed.

3. Are DLX1 associated downstream pathway genes involved in PSP?

4. Does under- and over-expression of DLX1AS affect DLX1 and MAPT expression?

5. Are there any changes in methylation-related proteins (readers and erasers)?

6. Genes such as ZIC5, METAP1D, SLIT1, PAX5, SP8 are all important in neurodevelopment, some in autism, and indeed methylation changes to DLX1 has been recently linked to immune-related neurodevelopmental disorder [Richetto et al (2017, Biological Psychiatry); should be included in discussion at least]. The authors could have used pyrosequencing to validate these genes (SLIT1 was done), as a suggestive neurodevelopmental theory of PSP seems to be the major conclusion of this study?

Reviewer #3 (Remarks to the Author):

Summary: Weber et al performed a genome-wide DNA methylation (DNAm) analysis of postmortem brain tissue comparing patients with Progressive Supranuclear Palsy (PSP) to unaffected controls. Some of the largest and most significant changes in DNAm levels were identified in DLX1. Subsequent transcriptional analyses of transcripts in DLX1 showed differential expression of an antisense transcript DLX1AS but not the full length/sense transcript, and there was no corresponding difference in protein levels.

While the QC and preprocessing of the DNAm data were rigorous, the differential methylation modeling was somewhat lacking. It was unclear the motivation for the Wilcoxon test used here, as linear modeling, which is standard in the field, is generally more powerful and can better prevent potential false positives due to the ability to adjust for confounders. For example, one potential confounder alluded to by the authors for this particular disorder is cellular composition, which can be estimated from existing cell sorted brain data and explored plus adjusted for in subsequent analysis [PMIDs: 24000956, 26619358, 23426267]. The authors should probably confirm that these dissections have similar composition estimates regardless of diagnosis, and perform differential methylation analyses that adjust for potential confounders (age, sex, batch, etc). Without these more rigorous differential methylation analyses, it is difficult to interpret the rest of the results of the paper, as the results and subsequent biological inference will likely change a lot.

Dear Dr. Trenkmann,

Below please find our point-by-point reply to the reviewers' comments on our paper "Epigenome-wide DNA methylation profiling in Progressive Supranuclear Palsy reveals major changes at *DLX1*"

Reviewer #1:

The manuscript by Weber et al is a genome-wide DNA methylation study in post-mortem brain tissue of individuals with PSP and controls.

*Clearly, the use of post-mortem brain tissue is a unique feature of this project and a major advantage in the study of neuropsychiatric disorders compared to the use of peripheral tissue. The manuscript is well written, concise and the finding of the *DLX1* antisense transcript expression influenced by DNA methylation is intriguing.*

However, there are several addressable concerns that limit my enthusiasm at this point:

Question 1: *Genotype: Although the authors test for known genetic variants in *MAPT* influencing the development of PSP, genome-wide genotyping and mQTL analysis should be performed. One first step could also be to check for SNPs that are known regulators of the CpG sites found in this study.*

Reply: To meet the reviewer's request, we have analyzed all SNPs in the present sample that we found associated with PSP in our previous

GWAS study (Höglinger et al., Nat. Genet. 43: 699-705 (2011)). We did not find any association between given SNPs and methylation in close proximity to these SNPs (± 50 kb) applying mQTL analysis.

The results are shown at the end of this letter (Rebuttal-Letter-Figure-1), and in the attachment (For editor Tables 1-5).

Interestingly, there appears to be some association between genotypes at *MOBP* and the degree of *DLX1* methylation when comparing the genotypes with the 265 significantly differentially methylated positions (“For editor Table 6”).

However, we prefer not to include these data in the manuscript since they will not withstand stringent corrections for multiple tests. Furthermore these findings of associated loci on different chromosomes would require in-depth functional studies which cannot be performed within the available time-frame.

Question 2: *Power analysis: The overall sample size is small, although I recognize the value of the brain samples and the difficulties to obtain larger number of samples. However, with 94 vs 72 there is essentially no power to detect small differences unless the authors can show otherwise. I tried to calculate the exact power, however, some information are missing from the methods incl the number of CpG that were tested etc. The authors may want to consider a better data reduction strategy to reduce the number of test to correct for multiplicity.*

Reply: We agree with the reviewer that our sample size does not provide sufficient power to detect small differences. We want to stress that we have followed the suggestion of reviewer #3 and have re-analyzed the entire dataset using state-of-the-art statistical approaches that are

commonly used in recent EWAS. As a result, the number of the originally described differentially methylated sites decreased. However, and most importantly, the methylation differences at *DLX1* became even more pronounced. We have replaced Figs.1 and 2 and the relevant supplementary tables and rewritten the text accordingly (see answer to reviewer 3). Thus, although not being sufficiently powered to detect small differences, the detected differences (265 dys-methylated sites, supplementary table 2), in particular at *DLX1/DLX1AS* are quite pronounced and real.

We added the following paragraph to the Materials and Methods section (subheading: **Array analysis**):

We calculated the exact statistical power for all 485577 probe sets on the microarray applying the function `pwr.t2n.test` in the `pwr` package of R statistical software. The parameters of this function are the number of patients and controls, i.e. $n=94$ and $n=71$, respectively, as well as the significance level (corrected $P<0.05$) and Cohen's d (effect size). The effect size is defined by the difference of the group means divided by the standard deviations of the two groups.

Given these parameters, a power of at least 80% is calculated for 14553 out of the total of 485577 probes. The statistical power within the subgroup of the 265 CpG sites significantly dys-methylated in PSP as compared to controls (corrected $P<0.05$ considering covariates) is at least 80% in 260, i.e. 98%.

Question 3: *Cell type specificity: The authors show desitometry and western blot data for DLX1. I was wondering if a deeper description of the DLX1 cells in the PSP tissue would be useful as an avenue for novel treatment approaches. Is the DLX1AS transcript expressed in all the DLX1 positive cells, what are other markers that define this population?*

Reply: We find a correlation between *DLX1* and *DLX1AS* expression in primary tissue from both PSP and controls (Rebuttal-Letter-Figure-2). Given that post mortem tissue was tested, no quantification is possible in different cell types. However, in neuroblastoma cell lines *DLX1AS* expression levels generally correlated with those of *DLX1*: SY5Y cells express lowest levels of both transcripts, IMR5 intermediate levels, and highest transcript levels were found in SH-EP cells. In one cell line, i.e. Ntera2-cells, however, *DLX1* expression was very low but that of *DLX1AS* was intermediate.

A more detailed description of the cells expressing *DLX1* on the APERIO slides was not possible. We could only manually assign the cell types with the highest *DLX1* expression (Supplementary Fig.7)

In Suppl. Figure 7 we show the detailed results of immunostaining using DLX1 antibodies:

Supplementary Figure 7 | Quantitative assessment of DLX1 staining of various cell types

Cell type assignment was done microscopically. Note highest expression of DLX1 in neurons of grey matter. There were no

significant differences in the relative proportion of DLX1-immunoreactive positive cells between PSP patients and controls.

Question 4: *Functional impact: The authors present gene expression, western blot and other data related to DLX1. I was wondering if a causal proof of differential methylation at the DLX1/DLX1AS locus using CRISPR/Cas in cell culture could firmly establish the link between DNA methylation and expression differences on the mRNA and protein level.*

Reply: We agree that CRISPR/Cas experiments could be informative. Unfortunately they could not be done within the available time-frame and will be a focus of future experiments. However, we performed numerous additional experiments in cell systems on the function of *DLX1/DLX1AS*.

For our experiments we used different cell lines. We added a paragraph with a description of the cell lines used to the Materials and Methods section (subheading: **Cell culture experiments**):

We used Ntera2, SH-EP cells, and strNPCs in our studies.

Ntera2 cells (pluripotent human embryonal carcinoma cells with characteristics of neuroepithelial precursor cells) were applied to overexpression experiments of DLX1 since intrinsic DLX1 mRNA expression is low (Rebuttal-Letter-Figure-3) . Ntera2 cells display neuronal characteristics (e.g. differentiation into neurons upon induction with retinoic acid), grow well under standard cell culture conditions and can be easily transfected.

SH-EP cells, a human neuroblastoma – derived cell line, were used in experiments involving DLX1AS overexpression. Endogenous

expression of DLX1 is increased as compared to other cell lines tested. DLX1AS was much less yet comparable in all cell lines tested. SH-EP cells are derived from neuronal tissue, grow well under standard cell culture conditions and can be easily transfected.

We used human striatal neuronal precursor cells (strNPCs) for siRNA experiments, since experimental conditions, in particular for MAPT/tau co-transfection experiments had already been established for these cells in our laboratory. Furthermore, these cells are commonly used as cultured cell models of neuronal cells: They are not transformed and thus lack the malignant phenotype of other cell lines.

Our findings are described in Results under the subheadings “Overexpression of *DLX1* and *DLX1AS* in Ntera2 and SH-EP cells” and “siRNA-mediated down-regulation of *DLX1* and *DLX1AS* in human striatal neuronal precursor cells (strNPCs)” and are presented in Fig.5:

Overexpression of *DLX1* and *DLX1AS* in Ntera2 and SH-EP cells

In order to study the function of DLX1 and DLX1AS we transfected Ntera2 and SH-EP cells using eukaryotic expression vector (pcDNA3.1-TOPO) containing either DLX1 or DLX1AS. Initial experiments had shown that untreated Ntera2 cells express less DLX1 than SH-EP cells (not shown). DLX1 was overexpressed 3-4 - fold in Ntera2 cells (Fig. 5a) and DLX1AS was overexpressed 100-120 - fold in SH-EP cells (Fig. 5b). We then proceeded to test the expression of known target genes of DLX1, i.e. GAD1²⁰, BRN3B²¹, and OLIG2²². All three genes were upregulated in cells

overexpressing DLX1 (Fig. 5a) and downregulated in cells overexpressing DLX1AS (Fig. 5b) Overexpression of DLX1 or DLX1AS did not affect expression of target genes GAD2 and GnRH (not shown).

We also tested expression of MAPT that had previously been shown to play an important role in the development of PSP^{4,5}. MAPT expression was reduced in cells overexpressing DLX1 (Fig. 5a) and increased in cells overexpressing DLX1AS (Fig. 5b).

siRNA-mediated down-regulation of DLX1 and DLX1AS in human striatal neuronal precursor cells (strNPCs)

In order to test whether a putative down-stream effect of DLX1/DLX1AS on MAPT also occurs in NPC lines derived from human fetal striatum (strNPC) we transfected these cells with siRNAs that target DLX1 and DLX1AS. As shown in fig. 5c knock-down of DLX1 resulted in significant upregulation of MAPT and of DLX1AS expression. Conversely, knock-down of DLX1AS resulted in downregulation of MAPT and in upregulation of DLX1.

We proceeded to test whether DLX1 affects Tau-dependent viability of strNPCs using the ATP firefly luciferase assay²³. strNPCs overexpressing either 3R or 4R Tau were co-transfected with siRNAs directed against either DLX1 or DLX1AS. As shown in fig. 5d, siRNA-mediated knock-down of DLX1 decreased cellular survival, in particular of cells overexpressing Tau. Conversely, knock-down of DLX1AS significantly increased survival of strNPCs (Fig. 5d).

Question 5: *Description of sample: The descriptions of the various post-mortem brain samples are insufficient (or basically not existing). There are no information on e.g. PMI, brain weight, medications, other diagnoses, hemisphere used, storage, RIN, ethnicity etc etc... I'm wondering if the authors even include the covariates in their methylations analyses? What covariates were included in the case-control design (not only for the methylation analyses but in general)? The methods did not mention gender and age, is this correct? Did the authors correct for batch effects (e.g. chip, bisulfite conversion?). A table with demographics is essential.*

Reply: We added all clinical information on patients and controls that was available. We now give the data in **Supplementary Table 1: Information on clinical and neuropathological findings in patients and controls**

Age and gender are given in this table. Both parameters were considered as co-variates in the revised analysis of the data (see above) We also give Braak and Thal stages, *TDP-43*, *APOE*, *MAPT* haplotype for each sample.

We now state in Materials and Methods (subheading "Array analysis"): In order to exclude batch effects we alternated hybridization of PSP and control samples.

Reviewer #2:

Weber and colleagues profile 94 PSP and 72 control prefrontal samples for methylation changes using 450K BeadChips. 621 sites showed >1% while only 48 showed >5% change. Most changes were associated with protein-coding genes. Pyrosequencing confirmed some candidate sites. The authors found >5% hypermethylation at the 3'UTR region of DLX1 that overlapped with exon 3 of DLX1AS whose transcript was altered in the disease samples, correlating with the change in methylation, and is proposed to alter DLX1 translation and contribute to PSP. The number of samples is appropriate, as are statistical methods.

Reply: We appreciate that this reviewer agrees that the sample size of our study is appropriate for significant statistical analysis.

Little is known regarding PSP at a molecular level, especially any involvement of lncRNAs, The study therefore is useful for the field and may instigate more functional studies. I have a few suggestions to strengthen the manuscript:

Question 1: *To confirm qPCR results, as opposed to using qPCR primers only from DLX1AS exon 1 (as this enhancer region may be independently transcribed) authors could use primer pairs from the common sequence of DLX1 and DLX1AS as well as unique DLX1 3'UTR sequence. The qPCR results should be identical if these two regions are co-transcribed.*

Reply: We tested several primers located within different exons or covering an exon-exon junction of DLX1AS to address this reviewer's question.

In agreement with prior experiments the SYBR-green based qPCR assays resulted in PCR products with more than one specific melting curve even after intensive optimization of the PCR conditions. These findings may be explained by different splice variants, but amplification of non-specific products cannot be excluded. However, two additional primer pairs that span several exons of *DLX1AS* gave some specific amplification products. The amount of these products correlated with that obtained with the highly specific primers from exon1 (Rebuttal-Letter-Figure-4).

We included the following paragraph in Materials and Methods (5th paragraph of subheading “Nucleic acids and PCR”):

We have used DNase treated polyA+ RNA in all RT-qPCR experiments. This excluded co-amplification of DNA that might have contaminated RNA preparations. This was particularly important for amplification of exon1-derived RNA of DLX1AS. Exon1 RNA of DLX1AS was analyzed since this exon was present in all alternative transcripts. However, we also tested two intron-spanning primer pairs of DLX1AS and found correlation of delta ct values with those of exon1-only amplified DLX1AS (data not shown).

Question 2: Direct binding of DLX1 to the predicted MAPT sites should be performed.

Reply: We performed three ChIP-qPCR experiments. However, we did not find any specific binding of DLX1 to the predicted *MAPT* sites.

In the first experiment ChIP-qPCR analyses from human neuronal precursor cells (immortalized LUHMES neurons and striatal fetal human neural precursor cells (NPCs)) with four different antibodies against endogenous DLX1 did not yield any enrichment of a number of candidate binding sites at the *MAPT* locus which were selected based on *in silico* analyses and chromatin marks from ENCODE datasets. However, we also did not find enrichment of previously defined DLX1 binding sites at the *GAD1*, *GAD2*, *DLX5/DLX6*, *GNRH*, *OLIG2* and *POU4F2* loci.

The second experiment with SH-EP cells (neuroblastoma cell line) using the same antibodies was negative.

In the third experiment V5-tagged DLX1 was ectopically expressed in HEK293 cells, in which the *MAPT* gene is expressed. The *MAPT* regions and the set of *bona fide* DLX1 target regions were not enriched by the V5 antibody.

A high-throughput ChIP-sequencing study with enrichment profiles for a large set of factors (Yan et al., ref. 45) used an antibody against DLX1 in chromatin preparations from the LoVo colon carcinoma cell line; no enrichment was found at or near the *MAPT* locus, suggesting indirect regulation.

We found one of the high-confidence binding sites from the study (*ACN9* locus) to be noticeably enriched by a different antibody against DLX1 in our SH-EP and LUHMES experiments, but we were concerned about the specificity of the ChIP assays due to the lack of enrichment of a regulatory region from a known DLX1 target gene. Importantly, a positive control IP against RNA polymerase II robustly and specifically enriched the transcription start sites of several loci in a cell-type specific manner. Therefore, we can presently neither confirm nor rule out regulation of *MAPT* by direct binding of DLX1 (Rebuttal-Letter-Figure-5).

Antibodies used for ChIP analysis:

Anti-RPB1 CTD Antibody, clone 8WG16, no. 664906, Biolegend

Anti-DLX-1 Antibody, clone L43/40, no. MABN467, Sigma

Anti-DLX-1 Antibody, clone 2H3, no. H00001745-M01, Abnova

Anti-DLX-1 Antibody, clone 4H7, no. H00001745-M14, Abnova

Anti-DLX-1 Antibody, no. HPA045884, Sigma

We summarize these findings in the Discussion of the revised manuscript: ***Currently, it remains unclear whether DLX1 influences MAPT expression directly or indirectly. Given that we could not conclusively show direct binding of DLX1 to the MAPT promoter (data not shown), the effect of DLX1 on MAPT appears to be more indirect (via the Wnt or the GABAergic interneuron-related network pathway). Furthermore, a high-throughput ChIP-sequencing study with enrichment profiles for a large set of factors⁴⁵ also used an antibody against DLX1 but did not find enrichment at or near the MAPT locus.***

Question 3: *Are DLX1 associated downstream pathway genes involved in PSP?*

Reply: Three genes downstream of *DLX1* (*GAT1*, *BRN3B*, *OLIG2*) were dysregulated in cell culture experiments. (See new **Fig. 5**).

Question 4: *Does under- and over-expression of DLX1AS affect DLX1 and MAPT expression?*

Reply: Yes, we found that overexpression of *DLX1* results in down-regulation of *MAPT*. Conversely, overexpression of *DLX1AS* results in down-regulation of *DLX1* and in upregulation of *MAPT* (this is shown in Fig. 5 and is described under the subheading “**Overexpression of *DLX1* and *DLX1AS* in *Ntera2* and *SH-EP* cells**”. Please also see our response to Reviewer #1 (above).

Question 5: Are there any changes in methylation-related proteins (readers and erasers)?

Reply: This is an important question. Since the observed methylation differences comprise both hyper- and hypo-methylation, it is unlikely that only readers or only erasers were responsible for the observed changes. We also did not find any differential methylation of DNMTs or TETs in our sample. However, analysis of such proteins is beyond the scope of this study which emphasizes the role of *DLX1* in PSP.

Question 6: Genes such as *ZIC5*, *METAP1D*, *SLIT1*, *PAX5*, *SP8* are all important in neurodevelopment, some in autism, and indeed methylation changes to *DLX1* has been recently linked to immune-related neurodevelopmental disorder [Richetto et al (2017, *Biological Psychiatry*); should be included in discussion at least]. The authors could have used pyrosequencing to validate these genes (*SLIT1* was done), as a suggestive neurodevelopmental theory of PSP seems to be the major conclusion of this study?

Reply: The new and most up-to-date data analysis did not reveal differential methylation of several previously assumed dys-methylated

genes (*ZIC5*, *PAX5*, *SP8*). Therefore we did not further analyze these genes. In addition to *DLX1*, pyrosequencing was performed of *DLX2*, *METAP1D*, *SLIT1*, *TRRAP*, *SCL15A3*. The findings confirmed the CHIP-results and the data is given in Supplementary Fig. 1:

Supplementary Figure 1 | Validation of differentially methylated genes.

*The differential methylation by BeadChip analysis was confirmed by pyrosequencing of the following genes: 1) DLX2 (5 CpG sites), METAP1D (3 CpG sites), SLIT1 (2 CpG sites), TRRAP (1 CpG site), SCL15A3 (2 CpG sites). The line in the middle of the box and whisker graph is the median (50th percentile). The box extends from the 25th to 75th percentile. The whiskers extend down to the lowest value and up to the highest. Statistics for each CpG site are given in a separate table under the graph. Welch's corrected t-test was used for data which are normally distributed in nearly all cases of CpG-control and CpG-PSP samples according to Kolmogorov-Smirnov. For data which are not normally distributed the Mann-Whitney test was used. *P<0.05, **P<0.01, ***P<0.001, n.s. = not significant.*

Reviewer #3:

Reviewer #3 (Remarks to the Author):

Summary: Weber et al performed a genome-wide DNA methylation (DNAm) analysis of postmortem brain tissue comparing patients with

Progressive Supranuclear Palsy (PSP) to unaffected controls. Some of the largest and most significant changes in DNAm levels were identified in DLX1. Subsequent transcriptional analyses of transcripts in DLX1 showed differential expression of an antisense transcript DLX1AS but not the full length/sense transcript, and there was no corresponding difference in protein levels.

Question 1: *While the QC and preprocessing of the DNAm data were rigorous, the differential methylation modeling was somewhat lacking. It was unclear the motivation for the Wilcoxon test used here, as linear modeling, which is standard in the field, is generally more powerful and can better prevent potential false positives due to the ability to adjust for confounders.*

Reply: We reanalyzed all data according to the reviewer's suggestions applying linear regression with correction for appropriate covariates. See also comment to reviewer #1.

In the revised version of our manuscript, we have re-done the statistical analysis of the 450K BeadChip data. We agree with the comments of reviewers 3 and 1 that use of a simple Wilcoxon test for the analysis of the differential DNA methylation does no longer represent the state of the art and that linear regression analyses are more powerful. Most importantly, linear regression analysis does allow the inclusion of covariates such as age and sex, which in the original analyses were only tested after the initial determination of the statistical significance. Similarly, over the last two years the processing of the array data has been improved by many groups and we chose to use an up-to-date pipeline for the analysis incorporating these results and filters (ChAMP),

which is widely used in the field. While there have been some changes to the gene lists due to slightly different pre-processing and - more importantly - the use of linear regression with age and sex as covariates, our overall findings on the changes in *DLX1* do not only hold, but are actually reinforced with more probes being significant after adjustment for multiple testing.

We describe this analysis in Materials and Methods:

Array analysis

The Infinium 450k array of Illumina Inc. (San Diego, CA) was used for methylation analysis. This array analyzes more than 485,000 CpG sites distributed over the entire genome. 200 ng of bisulfite-converted DNA were hybridized to the arrays according to the manufacturer's instructions. In order to exclude batch effects, we alternated hybridization of PSP and control samples. Arrays were scanned on an Illumina iScan platform at the Life & Brain Center (Bonn/Germany). Array data were evaluated, preprocessed and normalized using the ChAMP pipeline (V.2.8.1)^{50,51} with minor modifications. Raw array data were uploaded to the ChAMP pipeline using the minfi package^{52,53}

Probes with less than three measured beads or a detection P-value > 0.01 as well as probes interrogating CpGs that fall on or near to a SNP were removed based on recommendations by Zhou et al.⁵⁴. Similarly, probes aligning to multiple locations as defined by Nordlund et al.⁵⁵ were removed from the analysis. All samples had more than 99.3 % of valid probes, therefore no samples were removed from the analysis based on the quality of the data. The sex

of each sample was verified using an in-house R script based on the DNA methylation profile of the X and Y chromosomes. One single control sample was not in concordance with the assigned sex and was excluded from further analysis. Differences between probes due to Infl and Inll probe usage as well as between samples were normalized using BMIQ⁵⁶ on the 435 803 remaining probes. Differential methylation was determined using a modified version of the champ.dmp function with age and sex as covariates in a linear regression analysis based on the limma package⁵⁷ adjusting P-values for multiple testing with a Benjamini-Hochberg correction¹³. Differentially methylated CpGs were considered significant at adjusted P-values < 0.05.

The revised results are:

Differentially methylated sites in PSP

The genome-wide DNA methylation patterns of 94 PSP patients (72±5.3 years; 57% male, 43% female) were compared to 71 controls (76±7.9 years; 67% male, 33% female) without neurological or psychiatric diseases (Supplementary Table1). We studied prefrontal lobe tissue since it is consistently pathologically damaged in PSP, but less so than other brain regions³. Thus we detected disease-specific alterations and minimized a potential bias via a massive change in regional cellular composition due to severe neurodegeneration and gliosis.

Methylation differences at CpG sites between patients and controls were analyzed on 450 K BeadChips¹² at a Benjamini-Hochberg¹³ corrected level of significance of P<0.05.

Significant methylation differences were detected at 265 sites (227 hyper-, 38 hypo-methylated). Mean differences >5% were found at 32 of these sites (28 hyper-, 4 hypo-methylated) (Fig. 1a, Supplementary Table 2). Of the hypo-methylated sites, 68% were associated with protein-coding genes, 3% with genes for non-coding RNAs (miRNAs, lncRNAs, etc.), and 29% were located beyond 1.5 kb of genes. The respective percentages for hyper-methylated genes were 60%, 6%, and 34% (Fig. 1b).

The percentage of hypo-methylated CpG sites within gene bodies was 37% for hypo- and 26% for hyper-methylated sites (Fig. 1c). 24% of hypo- and 29% of hyper-methylated CpG sites were located within 5'UTRs (Fig. 1c). 8% hypo- and 2% hyper-methylated sites were within exon 1.

Among sites hypo-methylated in patients 50% were CpG islands and among hyper-methylated sites 30% represented CpG islands. 34% CpG sites were isolated ("open sea"¹²) and 25% of hyper-methylated sites were in "open sea" regions. The percentages of hyper-methylated CpG sites in "shelves" (2-4 kb from CpG island) was 3%, and that in "shores" (up to 2 kb from a CpG island¹²) was 13%. The corresponding values for hyper-methylated sites are 7% and 38% (Fig. 1d).

The circle plot was revised accordingly:

Chromosomal location of differentially methylated genes

Fig. 2 depicts the 139 genes with significant methylation differences between patients and controls and highlights the 9 genes with

differences >5% (see also Supplementary Table 2). Seven of these genes are hyper- and 2 are hypo-methylated.

Question 2: For example, one potential confounder alluded to by the authors for this particular disorder is cellular composition, which can be estimated from existing cell sorted brain data and explored plus adjusted for in subsequent analysis [PMIDs: 24000956, 26619358, 23426267]. The authors should probably confirm that these dissections have similar composition estimates regardless of diagnosis, and perform differential methylation analyses that adjust for potential confounders (age, sex, batch, etc). Without these more rigorous differential methylation analyses, it is difficult to interpret the rest of the results of the paper, as the results and subsequent biological inference will likely change a lot.

Reply: Suppl. Figures 5, 6, 7 address cell-type specificity (please see answer to reviewer 1 above). Cell-sorting could not be performed of our post mortem samples. We analyzed prefrontal lobe tissue since it is consistently pathologically damaged in PSP, yet, less so than in other brain regions. This allowed us to detect disease-specific alterations but minimized potential bias owing to a massive change in regional cellular composition. We state in Results (under subheading “Differentially methylated sites in PSP”):

We studied prefrontal lobe tissue since it is consistently pathologically damaged in PSP, but less so than other brain regions³. Thus we detected disease-specific alterations and minimized a potential bias via a massive change in regional cellular composition due to severe neurodegeneration and gliosis.

Gene ontology enrichment analysis did not reveal enrichment of specific cell types nor did it indicate an increase in cell death (apoptosis, senescence). We add under “***Functional analysis of differentially methylated genes***” ***No enrichment was found for genes involved in cell death and apoptosis.***

Age and sex were included in the main analysis (see above) and batch effects were avoided by alternating hybridization of samples (see above).

Description of changes in Figures 1-6 of the previously uploaded manuscript:

Figure 1 was changed according to the new data analysis. Figure 2 also takes into account the new data analysis. Furthermore, layout was altered for better understanding. In Figure 3 colors of the heat map were changed. Hypo-methylation is orange and hyper-methylation blue. This color assignment is now consistent in all relevant figures and tables. In Figure 4.e and 4.f labeling of the scale bars was removed. It is now described in the figure legend. Figure 5 is new and gives results of cell culture experiments. Pathway analysis (Figure 6) was re-done according to the new dataset of significantly dys-methylated genes. Supplemental material was also adjusted to the new data analysis.

Rebuttal-Letter-Figures

Figure_1

SNP allele distribution in PSP and controls

Rebuttal-Letter-Figure-1: Distribution of SNP-Genotypes of known GWAS SNPs relevant to PSP

GWAS analysis identified two SNPs at the *MAPT*, and 1 SNP each at *MOBP*, *EIF2AK3* and *STX6* that increase risk for PSP. All samples (n=94 PSP and n=71 controls) were genotyped for these five SNPs, i.e. rs242557, rs8070723, rs1768208, rs7571971 and rs1411478. Bar charts give the frequency of the genotypes in controls (grey bars) and PSP (red bars). Student's t-test was applied to the calculation of potential differences in genotype distribution using either a dominant (upper P-value) or a recessive (lower P-value) model of inheritance with respect to the variant- (non wild-type-) allele.

Figure_2

Rebuttal-Letter-Figure-2: Correlation of *DLX1* and *DLX1AS* mRNA expression

mRNA expression of *DLX1* and *DLX1AS*, shown as Delta-Ct-qPCR values correlates significantly within a) controls and b) PSP brain tissue specimens. The data does not distinguish between different cell types.

Figure_3

Rebuttal-Letter-Figure-3: *DLX1* and *DLX1AS* expression in different untreated cell lines (SY5Y, SH-EP, IMR5 and Ntera2).

We used *CYC1* and *EIF4A2* as housekeeping genes. a) qPCR Delta-Ct-values. *DLX1* expression is highest in SH-EP cells and lowest in Ntera2-cells. *DLX1AS* expression is also found highest in SH-EP cells but at a significantly lower expression level. b) Relative normalized expression (NE). NE is the relative quantity of the target (gene) normalized to the quantities of the reference targets (*CYC1* and *EIF4A2*). No single cell line was used as an internal control. Therefore the NE for *DLX1* and *DLX1AS* was scaled by dividing the expression level of each sample by the geometric mean level of expression of all samples. The software sets the average level of expression to a value of 1 and re-scales the expression levels for all samples.

Figure_4

Rebuttal-Letter-Figure-4: Comparison of different *DLX1AS* primer combinations.

Delta-Ct values of three different combinations of primers for *DLX1AS* qPCR correlate significantly in n=24 controls. mRNA isolation and

reverse transcription were performed as described in “Material and Methods” of the manuscript. Primers used for PCRs were:

qPCR1: “DLX1AS only ex. 1” (both primers in exon 1 (enhancer region))

upper-GATAGGAGGATGGGTCTG, lower-TGGACACACACTCTTTGC

qPCR2: “DLX1AS ex.1>ex.junction” (upper primer in exon 1, lower primer spanning exons 1 and 2)

upper-GATAGGAGGATGGGTCTG, lower-GGCTGAGAGGCCTTCTTC

qPCR3: “DLX1AS ex.1>ex.3” (upper primer in exon 1, lower primer in exon 3)

upper-TATTCTCCCAGCCAAAAGCTA, lower-
CCTCATCCGTCCGCTGTC

Pearson Correlation coefficient was calculated using R. P-values were calculated according to: *Cohen, J., Cohen, P., West, S.G., and Aiken, L.S. (2003). Applied Multiple Regression/Correlation Analysis for the Behavioral Sciences (3rd edition). Mahwah, NJ: Lawrence Earlbaum Associates.*

Figure_5

Rebuttal-Letter-Figure-5: Chromatin-Immunoprecipitation

ChIP-qPCR analyses in human neuronal precursor cells (LUHMES and striatal NPCs) with four different antibodies (L43, 2H3, 4H7, 045884) against endogenous DLX1 did not yield any enrichment of six candidate binding sites at and around the *MAPT* locus (-415,000; -2,000; -500; +9,000; +37,000; +215,000 bp relative to the transcription start site). The putative binding sites had been selected based on *in silico* analyses and chromatin marks from ENCODE datasets. However, DLX1 binding sites described at the *GAD1*, *GAD2*, *DLX5/DLX6*, *GNRH*, *OLIG2* and *POU4F2* loci were also not enriched. At the *ACN9* locus, one of the high-confidence binding sites detected with a different antibody against DLX1 by Yan et al. 2013, was enriched using the 2H3 antibody. However, we are concerned about the specificity of the ChIP assays due to the lack of enrichment of a regulatory region from a known DLX1 target gene. A positive control IP against the C-terminal domain of RNA polymerase II (RPB1 CTD), however, robustly and specifically enriched the *ACN9* site.

Legend to “For Editor Tables 1-5”: Analysis of methylation within 50 kb of DNA surrounding SNPs rs242557 (Table 1), rs8070723 (Table 2), rs1768208 (Table 3), rs7571971 (Table 4) and rs1411478 (Table 5). None of the CpGs probed on the 450K BeadChip 50 kb up or downstream of the five genotyped SNPs showed an association (after correction for multiple testing) with DNA methylation levels when performing cis-mQTL analysis using *MatrIXeQTL*. Data analysis: Genotype data on the 5 SNPs was investigated for potential methylation quantitative trait loci (mQTLs) using the normalised and quality controlled DNA methylation data for probes up to 50 kb up- and

downstream of each SNP (cis-mQTLs). mQTL analysis was performed with the MatrixEQTL package (Shabalin, A.A. Matrix eQTL: Ultra fast eQTL analysis via large matrix operations. *Bioinformatics* 28, no. 10 (2012): 1353-1358). eQTL (expression Quantitative Trait Loci) analysis using a linear model without any covariate and a FDR correction for multiple testing. T-statistics measures size of difference relative to variation in sample data. In our analyses T-statistics corresponds to the calculated difference in the degree of methylation between genotypes represented in units of standard error. The beta coefficient is the degree of change in methylation with respect to the different genotypes at a given SNP.

Legend to “For Editor Table 6”: SNP genotype - dependent methylation of the 265 significantly differentially methylated CpGs. Blue = hyper-methylated, orange = hypo-methylated, significant uncorrected P-values are marked red.

Tables are attached as separate Excel files.

Reviewer #1 (Remarks to the Author):

Thank you for the opportunity to review the revised version of the manuscript titled "Epigenome wide DNA methylation in Progressive Supranuclear Palsy reveals major changes at DLX1"

The authors responded to my questions, however, concerns remain:

Genotype: The authors tested for effect of GWAS-derived SNPs on DNA methylation and found no association. I still have reservations on the approach to focus on GWAS derived SNPs only, in particular when the 2011 GWAS was small (the authors tested essentially for 5 SNPs, DLX1 was not included as it did not show up in the GWAS). Why not genotyping all individuals from this study here genome-wide and test for genetic effects on DNA methylation? DNAm is strongly driven by genotype and it is more likely that DNAm differences are due to 1) genotype or 2) cell type composition. The authors did not exclude (both) confounders sufficiently. In addition, there are databases available with larger sample sizes that would enable the authors to test for genetic effects on DNAm (<https://epigenetics.essex.ac.uk/mQTL/>)

Power: I'm surprised to see enough power with N=94 vs N=71, which is in contrast to the published literature and recommendations on EWAS. I am not a bioinformatician/statistician and would like to refer this question to an appropriate consultant. I would be curious what effect size forms the basis of these assumptions. Unfortunately, the authors do not provide enough information to assess the statement that power is sufficient. Despite the strong data on the underlying biology of DLX1 presented here, I have concerns that the dataset that lead to this candidate finding is inappropriate.

Cell type composition: This is a major concern and mirrors similar comments from reviewer #3. Although the authors state that there were no significant differences between PSP and controls in terms of cell type proportions using immunostaining, Suppl Fig 7 clearly show strong differences for example in grey matter neurons (~65% of DLX1 pos cells were neurons in controls compared to 45% in PSP). The notion that prefrontal lobe tissue is less damaged in PSP does not exclude that observed differences are derived from subtle cell type composition changes. EWAS in general is likely confounded by cell type changes that are not accounted for especially with results around 5-10% DNAm differences (<https://www.nature.com/articles/nrg.2017.32>). Why not using bioinformatic tools to account for cell type composition and all unknown or unaccounted confounders? This would also allow to control for covariates not included here such as PMI, medications etc. I'm concerned that the authors do not include relevant covariates. For example, a simple group comparison of the sparse clinical data given in Suppl Table 1 shows significant differences between PSP and controls in Braak stages and PMI. How did the authors control for medication? Smoking? Unknown factors?

Reviewer #2 (Remarks to the Author):

The authors have adequately addressed my comments.

Reviewer #3 (Remarks to the Author):

Summary: Weber et al largely addressed many of the concerns raised by the reviewers, including my comments. However, I still have concerns about the potential role of cellular composition in the results, which was also raised by Reviewer 1.

1. Reviewer 1, Question 3. The authors should examine single cell/nuclei datasets to determine the cell type specificity of the correlation between the sense (DLX1) and antisense (DLX1-AS) transcripts. There are many publicly available datasets including Darmanis et al PNAS 2015 [PMID: 26060301] and Lake et al, Science 2016 (which is only neuronal, PMID: 27339989) plus and a recently updated Nat Biotech 2017 [PMID: 29227469].

2. Reviewer 3, Question 2. The authors misunderstood my suggestion. There are statistical approaches to estimate the relative proportions of cell types (neurons and glia) that do not actually require sorting nuclei from postmortem tissue. The authors should use these tools to estimate cellular composition in their data and determine if the proportion of neurons or glia is indeed a confounder. The previous query again was: "For example, one potential confounder alluded to by the authors for this particular disorder is cellular composition, which can be estimated from existing cell sorted brain data and explored plus adjusted for in subsequent analysis [PMIDs: 24000956, 26619358, 23426267]. The authors should probably confirm that these dissections have similar composition estimates regardless of diagnosis, and perform differential methylation analyses that adjust for potential confounders (age, sex, batch, etc)." It does appear that the CpGs most differentially methylated for PSP were differentially methylated comparing neurons and glia from sorted samples [PMID: 23426267, via <https://bioconductor.org/packages/release/data/experiment/html/FlowSorted.DLPFC.450k.html>] with $p = 2.2e-10$. More importantly, this CpG is more highly expressed in glia, which would be predicted to be more abundant in PSP (due to loss of dopamine neurons) and this CpG was more highly methylated in patients than controls (see attached plot). Therefore, the authors should explore the potential role of cell type confounding in their genome-wide analyses.

Response to Reviewers' comments:

Reviewer #1 (Remarks to the Author):

Thank you for the opportunity to review the revised version of the manuscript titled "Epigenome wide DNA methylation in Progressive Supranuclear Palsy reveals major changes at DLX1"

The authors responded to my questions, however, concerns remain:

Question: *Genotype: The authors tested for effect of GWAS-derived SNPs on DNA methylation and found no association. I still have reservations on the approach to focus on GWAS derived SNPs only, in particular when the 2011 GWAS was small (the authors tested essentially for 5 SNPs, DLX1 was not included as it did not show up in the GWAS). Why not genotyping all individuals from this study here genome-wide and test for genetic effects on DNA methylation? DNAm is strongly driven by genotype and it is more likely that DNAm differences are due to 1) genotype or 2) cell type composition. The authors did not exclude (both) confounders sufficiently. In addition, there are databases available with larger sample sizes that would enable the authors to test for genetic effects on DNAm (<https://epigenetics.essex.ac.uk/mQTL>)*

Answer: We thank the reviewer for his/her efforts and suggestions, which have helped to further strengthen our data. The following additional observations mainly confirm the previous findings.

Concerning this reviewer's concerns on GWAS, the original 2011 GWAS comprised 2165 cases and 6807 controls (Höglinger et al., Nature Genetics 2011). Therefore the 5 SNPs identified at genome-wide significance in this GWAS study are based on a very solid foundation. These 5 SNPs are still considered the only reliable genetic variation that predisposes to PSP and were therefore chosen to study the effect of genetic variation on DNA methylation. Given the findings on genetic variation obtained in our previous large GWAS study (that included several thousand individuals), it is unlikely that genome-wide genotyping of the 94 PSP patients and the 71 controls of the current EWAS study could add more precise information on the genetic variation in PSP.

Following the recommendation of this reviewer and his/her well-founded concerns about the influence of genetic variation on the level of DNA methylation, we accessed the data of all significant mQTLs detected in adult prefrontal cortex in the study the reviewer referred to (Hannon et al., Nat Neurosci. 2016;19:48-54). We compared the methylation data of this study to our list of significant associations of DNA methylation with PSP, corrected for sex, age and non-neuronal cell content (see below). Only three CpGs of our list have previously been identified to be influenced by genetic variation. These are cg01378667, cg03325535, cg10318222 in the promoter region of *GABRA5*, all of which are associated in cis with rs7496866 located in the same region. Therefore we removed these three CpGs from our list of

significant associations. We can thus be confident that our findings represent true associations with high probability and are not mediated by genetic variations.

Changes in the manuscript:

The methylation data analyzed under consideration of the below described covariates and CpGs associated with genetic variations removed are shown in Supplementary Table 2. Figures 1, 2 and 3 have been changed according to the new results. All numbers and percentages have been revised in text and figures according to the new analysis. Supplementary Table 3 contains new *P*-values with respect to the novel corrections. Furthermore, the enrichment analysis and the pathway analyses have been adapted to the changes of the list of differentially methylated genes.

We added the following paragraph to the Results section (subheading: **Differentially methylated sites in PSP**):

Significant CpGs previously shown to be influenced by genetic variants (mQTLs) in adult prefrontal cortex¹⁵ were not included in further analyses. However, only three CpGs were found to match this criterion, i.e. cg01378667, cg03325535, cg10318222 in the promoter region of GABRA5, all of which are associated in cis with rs7496866 located in the same region.

Question: *Power: I'm surprised to see enough power with N=94 vs N=71, which is in contrast to the published literature and recommendations on EWAS. I am not a bioinformatician/statistician and would like to refer this question to an appropriate consultant. I would be curious what effect size forms the basis of these assumptions. Unfortunately, the authors do not provide enough information to assess the statement that power is sufficient. Despite the strong data on the underlying biology of DLX1 presented here, I have concerns that the dataset that lead to this candidate finding is inappropriate.*

Answer: The power of a hypothesis test is the probability that the test correctly rejects the null hypothesis (H0) when a specific alternative hypothesis (H1) is true. It is not possible to calculate the exact power for a microarray / EWAS study. The reason is that the parameters (μ_0 and μ_1) of the t-test have to be fixed when calculating the exact power. Therefore, the exact power can only be computed for one t-test at a time. In case of microarray / EWAS studies hundreds of thousands of t-tests and hundreds of thousands of effect sizes are determined. However, it is possible to estimate the power for a microarray / EWAS study by simulations (Tsai et al., Int J Epidemiol. 2015;44:1429-1441; Saffari et al., Genet Epidemiol. 2018;42:20-33). Tsai et al. applied power estimations to the same microarray / EWAS platforms as in our study (i.e. microarray based 450K Illumina DNA methylation platform). Their data indicates that 96 cases and 96 controls would be required to have 80% power to detect an 11% difference in DNA methylation at genome-wide significance, yet these studies were done in different biological systems (e.g. cancer, diabetes, smoking and blood), which may influence the distribution and variability of DNA methylation as compared to our biological context (neurodegeneration).

Therefore, we computed the exact statistical power for each of the 485577 probe sets on the microarray based on the data obtained in our sample by applying the function `pwr.t2n.test` of the `pwr` package of R statistical software. The parameters of this function are the number of patients and controls, i.e. $n=94$ and $n=71$, respectively, as well as the significance level of 0.05 and Cohen's d (effect size). The effect size is defined by the difference between the means of the group divided by the pooled standard deviations of the two groups. Given these parameters, a power of at least 80% is calculated for 14553 out of the total of 485577 probes. Taking into account the covariates gender and age in a generalized linear model most of the CpG sites which were significantly differentially methylated in PSP as compared to controls showed a power of greater than 80% to detect DNA methylation differences of distinctly less than 11%.

(see Rebuttal Table1 "power_calculation_of_all_probes.xlsx" column "part of the linear regression model using covariates", column "difference of means" and column "power" of the revised ms.).

Thus our study was sufficiently powered to detect the top specifically differentially methylated CpG sites.

Specifically, our sample allowed detection of significant (P (corrected) < 0.05) methylation differences at 717 CpG sites. Independent methods confirmed these differences at selected sites and one CpG cluster (*DLX1*) was further validated by functional assays.

We agree with the referee that there might be additional true methylation differences, for which we falsely rejected the null hypothesis (i.e. did not identify them). For this purpose, larger case numbers will indeed be helpful in future studies. However, this limitation does not question the methylation differences which we identified in the present sample.

Question: *Cell type composition: This is a major concern and mirrors similar comments from reviewer #3. Although the authors state that there were no significant differences between PSP and controls in terms of cell type proportions using immunostaining, Suppl Fig 7 clearly show strong differences for example in grey matter neurons (~65% of DLX1 pos cells were neurons in controls compared to 45% in PSP). The notion that prefrontal lobe tissue is less damaged in PSP does not exclude that observed differences are derived from subtle cell type composition changes. EWAS in general is likely confounded by cell type changes that are not accounted for especially with results around 5-10% DNAm differences (<https://www.nature.com/articles/nrg.2017.32>). Why not using bioinformatic tools to account for cell type composition and all unknown or unaccounted confounders? This would also allow to control for covariates not included here such as PMI, medications etc. I'm concerned that the authors do not include relevant covariates. For example, a simple group comparison of the sparse clinical data given in Suppl Table 1 shows significant differences between PSP and controls in Braak stages and PMI. How did the authors control for medication? Smoking? Unknown factors?*

Answer: We thank the reviewer for insisting on this important point that was also raised by reviewer 3. We have now used available reference data for neuronal tissue (Guintivano et al., Epigenetics. 2013;8:290-302) and implemented this function available in the minfi package in our processing pipeline (CHAMP). We first estimated neuronal and non-neuronal cell contents in our samples and present the results in the novel supplementary Figure 1 (see also Rebuttal Figure 1), showing a slight loss of neurons in the PSP patients as compared to controls. However, this small neuronal loss was not statistically significant.

To perform the primary analyses of methylation differences between cases and controls properly, we included the proportion of neuronal and non-neuronal cells as a covariable in the linear regression model. Interestingly, this nearly tripled the number of CpGs associated with PSP after correction for multiple testing (before 265, now 717 CpGs) (Supplementary Table 2).

This indicates that the cell-type composition masked some of the significant findings. Examination of the p-value distribution (QQ-plot) did not show any evidence of a *P*-value inflation or any other strong unknown confounders associated with cell-type composition (Rebuttal Figure 2).

Novel findings also included additional differentially methylated sites within *DLX1* and further corroborate our main observation. This further demonstrates that differential methylations within *DLX1* are the main epigenetic changes in adult pre-frontal cortex of PSP patients. Further analyses in larger cohorts might detect additional changes, but will not affect the relevance of the major findings presented in this manuscript.

Changes in the manuscript:

The corrected methylation data are displayed in Supplementary Table 2. Figures 1, 2 and 3 have been changed according to the new results (marked yellow in the manuscript). All numbers and percentage values have been corrected in text and figures according to the new analysis. Supplementary Table 3 contains new *P*-values with respect to the improved corrections. Furthermore, the new data have been incorporated in the enrichment and the pathway analysis.

We added the following paragraphs to “Results”
(subheading: **Differentially methylated sites in PSP**):

We estimated the amount of neuronal and non-neuronal cells in our samples as described by Guintivano et al.¹² The percentage of neurons in PSP patients (median 36.1% of cells) did not significantly differ from the proportion of neuronal cells in controls (median 38.0% of cells) (Wilcoxon Test, $P=0.31$) (Supplementary Fig. 1).

Methylation differences at CpG sites between patients and controls were analyzed on 450 K BeadChips¹³ applying a linear regression model with age, sex and non-neuronal cell content as covariates at a Benjamini-Hochberg¹⁴ corrected level of significance of $P<0.05$.

After these corrections, significant methylation differences were detected at 717 sites (627 hyper-, 90 hypomethylated).

We changed the following paragraph in “Results”
(subheading: **Pronounced hypermethylation of *DLX1***):

Differential methylation of $\geq 5\%$ was only detected at a few CpG sites in a small number of genes in PSP (Fig. 2, Supplementary Table 2). Most pronounced hypermethylation was detected at a region of chromosome 2 that includes the gene *DLX1* (Distal-Less Homeobox 1). Many sites of *DLX1*, mainly within its 3'UTR (Fig. 3a), were hypermethylated by $\geq 5\%$, as shown in a representative heat-map of 11 sites.

We changed the following paragraph in “Results”
(subheading: **Functional analysis of differentially methylated genes**):

We performed in silico functional analyses of 375 different annotated genes that are represented by 451 (out of a total of 717) differentially methylated CpG sites on the Illumina 450 kb chip (Supplementary Table 2). Applying the Pathway Studio software and Fisher's exact test we searched for enrichment of the Gene Ontology (GO) category “biological process” (Supplementary Table 4). *P* values were corrected for multiple testing according to Benjamini & Hochberg¹⁴. The top 20 highly significantly enriched categories include important functions pertinent to *DLX1* and *DLX2*, i.e. “anatomical structure development” (GO ID 48856), “regulation of signaling” (GO ID 23051), “cell fate commitment” (GO ID 45165), “positive regulation of transcription” (GO ID 45893) and the “cell surface receptor signaling pathway” (GO ID 7166) All 18 genes assigned to the term “Wnt signaling pathway” (GO ID 16055), which show a significantly corrected *P* value of 6.57×10^{-04} are part of the “cell surface receptor signaling pathway” (Supplementary Table 4). Particularly many neuronal functions and pathways are distributed across the list of significantly enriched categories, i.e. “neuron fate commitment” (GO ID 48663), “cerebral cortex GABAergic interneuron fate commitment” (GO ID 28193), and “negative regulation of neurogenesis” (GO ID 50768) (all of which include *DLX1* and *DLX2*, Supplementary Table 4).

We changed the following paragraph in “Results”
(subheading: **Pathway analysis of differentially methylated genes**):

Pathway analysis revealed interdependence of several genes found to be differentially methylated in PSP (Fig. 6, Supplementary Table 5). Based on literature mining we propose two main possible pathways linking *DLX1* and *MAPT*:

(1) Activation of *MAPT*-encoded Tau protein via the Wnt signaling pathway: This notion is supported by the finding that *DLX2* bound to *Necdin* activates the *WNT1* promoter²⁹. In PSP patients, *DLX1* and *DLX2* are highly ($\geq 5\%$) and the WNT ligand family members *WNT10A*, *WNT8b* are distinctly ($>3\%$) hypermethylated. Several additional differentially methylated genes

(methylation differences >1%) are members of the WNT signaling pathway as well (Supplementary Table 4). Furthermore, WNT signaling appears to affect Tau phosphorylation in Alzheimer's disease³⁰⁻³².

(2) Tau phosphorylation via GABA(A) receptors: DLX1, DLX2 and GABA(A) receptors (encoded by the differentially methylated genes *GABRA5*, *GABRB3* and *GABRD*) are members of the GABAergic interneuron-related network in humans³³. Within this network DLX1/DLX2 regulate GABA synthesis²⁴. Expression changes of *DLX1/DLX2* can alter activation of GABA(A). GABA(A) receptors in turn play an important role in Tau phosphorylation. This observation is consistent with the notion that Tau is affected via DLX1/DLX2 – GABA(A) in PSP as well³⁴.

Reviewer #2 (Remarks to the Author):

The authors have adequately addressed my comments.

Reviewer #3 (Remarks to the Author):

Summary: *Weber et al largely addressed many of the concerns raised by the reviewers, including my comments. However, I still have concerns about the potential role of cellular composition in the results, which was also raised by Reviewer 1.*

Question: *Reviewer 1, Question 3. The authors should examine single cell/nuclei datasets to determine the cell type specificity of the correlation between the sense (DLX1) and antisense (DLX1-AS) transcripts. There are many publicly available datasets including Darmanis et al PNAS 2015 [PMID: 26060301] and Lake et al, Science 2016 (which is only neuronal, PMID: 27339989) plus and a recently updated Nat Biotech 2017 [PMID: 29227469].*

Answer: Data from sorted and pooled cells from mouse brain tissue show that both, *Dlx1* and *Dlx1as* transcripts are expressed mainly in neuronal cells²² (Rebuttal Figure 4a). The file `barreslab_rnaseq.xlsx` was downloaded from https://web.stanford.edu/group/barres_lab/brain_rnaseq.html and the corresponding data were represented as a bar plot.

In order to find out whether both transcripts are also preferentially expressed in neuronal cells of human brains, we analyzed the dataset of Darmanis et al.²³ (GEO: GSE67835). This analysis demonstrated that *DLX1* and *DLX1AS* are mainly expressed in neurons (Rebuttal Figure 4b and 4c), and that neurons express either *DLX1* or *DLX1AS*, but only rarely (2.3% of neurons) express both *DLX1* and *DLX1AS* (Rebuttal Figure 4d).

Specifically, we downloaded data on single cells from GEO and processed them according to the recommendations detailed in : https://www.ncbi.nlm.nih.gov/geo/query/acc.cgi?link_type=NCBIGEO&access_num=GSE67835&acc=GSE67835

We applied the program Prinseq (-min_len 30) to remove very short non-specific reads. Prinseq also trimmed both ends of the reads in order to eliminate 5' duplicates (-trim_left 10) and to remove low quality 3' ends (-trim_qual_right 25). Furthermore Prinseq filtered reads of low complexity (-lc_method entropy \-lc_threshold 65). The program FASTQC was used to identify sequences that are overrepresented (adapter) in order to exclude them from further analysis. We used the Prinseq tool to remove orphan pairs less than 30bp in length followed by removal of nextera adapters using Trim Galore (--stringency 1). Reads were aligned to the hg19 genome with STAR using the following options (-outFilterType BySJout \-outFilterMultimapNmax 20 \-alignSJoverhangMin 8 \-alignSJBoverhangMin 1 \-outFilterMismatchNmax 999 \-outFilterMismatchNoverLmax 0.04 \-alignIntronMin 20 \-alignIntronMax 1000000 \-alignMatesGapMax 1000000 \-outSAMstrandField

intronMotif). Aligned reads were converted to counts for each gene using HTSeq (-m intersection-nonempty -s no). The Ensembl General Feature Format (GTF) annotation file necessary for HTSeq was extended by *DLX1AS* splice variants represented by the following Genbank identifier KU179668.1, KU179669.1, KU179670.1, KU179671.1 and KU179672.1. For all analyses Genome_build hg19 was used. Finally, counts were converted into FPKM (Fragments per kilobase of transcript sequence per million mapped fragments) values.

Out of 466 available single cell datasets we selected 251 datasets that represent specific cortical cell types [astrocytes, microglia, oligodendrocytes, endothelia, neurons, oligodendrocyte precursor cells (OPC)]

see <https://www.ncbi.nlm.nih.gov/Traces/study/?acc=SRP057196>

Hybrid cells and fetal quiescent cells were not considered.

We could not analyse the two additional datasets mentioned by the reviewer for the following reasons:

- The dataset of Lake et al. (Nat Biotechnol. 2018;36(1):70-80 [PMID: 29227469] (GEO: GSE97942) could not be analyzed since it requires polyA+ RNA as input. Given that *DLX1AS* is a lncRNA which lacks a poly A tail data on *DLX1AS* expression cannot be retrieved from this data base.
- The dataset of Lake et al. (Science. 2016;352(6293):1586-90 [PMID: 27339989] could also not be used since application at the NCBI db GaP and JAAMHDAC (NIH/NIDA) turned out to be complicated and time consuming and, although we have initiated this process, access could not be obtained within the time frame given for this resubmission. Moreover it turned out, that the dataset contains data from neuronal cells only, so a comparison of *DLX1/DLX1AS* expression in neuronal vs. non-neuronal cells might not be possible.

Given that the results of analysis of the Darmanis dataset were clear-cut, however, analysis of additional datasets is unlikely to give different results.

Changes in the manuscript:

We added the following paragraph to "Results":

(subheading: **Single cell analysis from healthy human cortex**):

Dlx1 and Dlx1as are almost exclusively expressed in neuronal cells in the mouse²² (Supplementary Fig. 4a). Using RNA sequencing data in single cells from healthy human cortex published by Darmanis and colleagues²³ we quantified the reads of *DLX1* and *DLX1AS* transcripts. Consistent with the mouse data, we found *DLX1* and *DLX1AS* expression mainly in neurons (Supplementary Fig. 4b and 4c). Expression of *DLX1* was detected in 26.15% of neurons (Supplementary Fig. 4d), as compared to 1.65% of all non-neuronal cells (not shown). The corresponding results for *DLX1AS* were 15.39% and 4.96% (not shown).

While in most single neurons either DLX1, DLX1AS or none of both were detected, joint expression of DLX1 and DLX1AS was observed in 2.31% of neurons only (Supplementary Fig. 4d).

We added the following paragraph to “Discussion”:

This is consistent with our observation of mainly neuronal expression of DLX1 in single cells, confirming previously described neuronal specificity of Dlx1 expression. Similarly, Dlx1as and DLX1AS are also predominantly expressed in neurons in both mouse and human. Interestingly, individual neurons of healthy human cerebral cortex expressed either DLX1 or DLX1AS, but only very few neurons expressed both DLX1 and DLX1AS, suggesting that the expression of DLX1 and DLX1AS are regulated in opposite manner (Supplementary Fig. 4). This interpretation would be consistent with our cell culture experiments showing increased DLX1AS expression upon DLX1 silencing and vice versa.

We added Rebuttal Figure 3 as new Supplementary Figure 4 in the manuscript. The subsequent numbering of the supplementary figures was adapted.

Question: Reviewer 3, Question 2. *The authors misunderstood my suggestion. There are statistical approaches to estimate the relative proportions of cell types (neurons and glia) that do not actually require sorting nuclei from postmortem tissue. The authors should use these tools to estimate cellular composition in their data and determine if the proportion of neurons or glia is indeed a confounder. The previous query again was: “For example, one potential confounder alluded to by the authors for this particular disorder is cellular composition, which can be estimated from existing cell sorted brain data and explored plus adjusted for in subsequent analysis [PMIDs: 24000956, 26619358, 23426267]. The authors should probably confirm that these dissections have similar composition estimates regardless of diagnosis, and perform differential methylation analyses that adjust for potential confounders (age, sex, batch, etc).” It does appear that the CpGs most differentially methylated for PSP were differentially methylated comparing neurons and glia from sorted samples [PMID: 23426267, via <https://bioconductor.org/packages/release/data/experiment/html/FlowSorted.DLPFC.450k.html>] with $p = 2.2e-10$. More importantly, this CpG is more highly expressed in glia, which would be predicted to be more abundant in PSP (due to loss of dopamine neurons) and this CpG was more highly methylated in patients than controls (see attached plot). Therefore, the authors should explore the potential role of cell type confounding in their genome-wide analyses.*

Answer: As detailed above (response to reviewer 1, question 3: Cell type composition) we estimated the relative proportion of neurons vs. non-neuronal cells in brains of PSP patients and controls. We did not find a statistically significant difference in the composition of these cell types, in particular of neuronal cells, between patients and controls.

Nevertheless we corrected for the differences in cellular composition as requested by reviewers 1 and 3. This correction is now taken into account in Supplementary Table 2 and in all pertinent figures and tables (see reply to reviewer 1, question 3)

Changes in the manuscript:

In addition to the changes made in response to the reviewers (above) we added the following paragraphs to Materials and Methods:

Array analysis: Specifically, raw array data were uploaded to the ChAMP pipeline using the minfi option⁵⁷. Relative proportions of neuronal and non-neuronal cells in each sample were estimated based on the raw data applying the compositeCellType="DL PFC" option of estimateCellCounts function of the R Bioconductor minfi package⁵⁸ ... *P*-values < 0.05. To account for potential confounding owing to genetic variation, significant probes were filtered for significant mQTLs in adult prefrontal cortex using previously published data, which removed three CpGs all of which associated with *GABRA5* (cg01378667, cg03325535, cg10318222) as defined by a SNP in close proximity to the gene (rs7496866)¹⁵.

Literature mining and pathway analysis: Using Network Builder of the Pathway Studio software (Elsevier) version 12.0.1.5 we generated a literature-derived network that was based on text mining for direct interactions between significantly differentially methylated input genes. The input dataset comprised genes adjacent to the 717 genomic positions (CpG sites) found to significantly differ in methylation between PSP patients and controls. We only analyzed those genes by Network Builder that are known to be expressed in the telencephalon and in interneurons. The GeneRanker program (Genomatix) was used for analysis of association of differentially methylated genes in tissues. The literature-derived network was curated and extended by manual literature searches as well as by in silico prediction of transcription factor binding sites applying the MatInspector program (Genomatix). The complete set of differentially methylated genes was further explored using the Pathway Studio software for enrichment in the category "biological process" of the Gene Ontology by applying Fisher's exact test. *P* values were corrected for multiple testing by Benjamini & Hochberg¹⁴.

Single cell analysis: We downloaded data on single cells from GEO and processed them according to the recommendations detailed in https://www.ncbi.nlm.nih.gov/geo/query/acc.cgi?link_type=NCBIGEO&access_num=GSE67835&acc=GSE67835²²

We applied the program Prinseq (-min_len 30) to remove very short non-specific reads. Prinseq also trimmed both ends of the reads in order to eliminate 5' duplicates (-trim_left 10) and to remove low quality 3' ends (-trim_qual_right 25). Furthermore Prinseq filtered reads of low complexity (-lc_method entropy -lc_threshold 65). The program FASTQC was used to identify sequences that are overrepresented (adapter) in order to exclude them from further analysis. We used the Prinseq tool to remove orphan

pairs less than 30bp in length followed by removal of nextera adapters using Trim Galore (--stringency 1).

following options (-outFilterType BySJout \--outFilterMultimapNmax 20 \--alignSJoverhangMin 8 \--alignSJDBoverhangMin 1 \--outFilterMismatchNmax 999 \--outFilterMismatchNoverLmax 0.04 \--alignIntronMin 20 \--alignIntronMax 1000000 \--alignMatesGapMax 1000000 \--outSAMstrandField intronMotif).

Aligned reads were converted to counts for each gene using HTSeq (-m intersection-nonempty \-s no). The human Ensembl General Feature Format (GTF) annotation file (version 2013-09) necessary for HTSeq was extended by DLX1AS splice variants represented by the following Genbank identifier KU179668.1, KU179669.1, KU179670.1, KU179671.1 and KU179672.1. For all analyses Genome_build hg19 was used. Finally, counts were converted to FPKM (Fragments per kilobase of transcript sequence per million mapped fragments) values.

Out of 466 available single cell datasets we selected 251 datasets that represent specific cortical cell types (astrocytes, microglia, oligodendrocytes, endothelia, neurons, oligodendrocyte precursor cells, see <https://www.ncbi.nlm.nih.gov/Traces/study/?acc=SRP057196>). Hybrid cells and fetal quiescent cells were not considered.

Additional changes:

New literature was added: References 14, 21, 22 and 25.

The Acknowledgment section was updated.

According to the Data availability statements and data citations policy we added a separate paragraph at the end of the Methods section:

Data availability

Normalized and raw BeadChipArray data have been deposited in NCBI- Gene Expression Omnibus (GEO) with the accession code GSE75704 (<https://www.ncbi.nlm.nih.gov/geo/query/acc.cgi?acc=GSE75704>).

DLX1AS transcript variants have been uploaded to NCBI under accession numbers KU179668, KU179669, KU179670, KU179671 and KU179672

Literature

12. Guintivano J, Aryee MJ, Kaminsky ZA. A cell epigenotype specific model for the correction of brain cellular heterogeneity bias and its application to age, brain region and major depression. *Epigenetics*, 3, 290-302 (2013)

15. Hannon, E. et al. Methylation QTLs in the developing brain and their enrichment in schizophrenia risk loci. *Nat Neurosci.* 19, 48-54 (2016)

22. Zhang, Y. et al. An RNA-sequencing transcriptome and splicing database of glia, neurons, and vascular cells of the cerebral cortex. *J Neurosci.* 34, 11929-11947 (2014)

23. Darmanis, S. et al. A survey of human brain transcriptome diversity at the single cell level. *Proc Natl Acad Sci U S A.* 112, 7285-7290 (2015)

26. Givens, M.L. et al. Developmental regulation of gonadotropin-releasing hormone gene expression by the MSX and DLX homeodomain protein families. *J Biol Chem.* 19, 19156-65 (2005)

We thank the reviewers for their thoughtful comments. By answering all their questions we think the manuscript has further improved greatly and we hope that our work can now be accepted for publication in Nature Communications.

With kind regards,

Ulrich Müller, Axel Weber, Günter Höglinger

Rebuttal Figures

Rebuttal Figure 1

Rebuttal Figure 1:

Proportion of neuronal and non-neuronal (glial) cells in PSP and control brains as estimated with the estimate CellCounts function from the mini Bioconductor package using dorsolateral prefrontal cortex (DLPFC) as composite cell type. Although there is - as expected- a slight decrease in neuronal cell content PSP brains (n=94) compared to control brains (n=71), the difference did not reach statistical significance (Wilcoxon Test, $P=0.31$). The line in the middle of the box and whisker graph represents the median (50th percentile). The box extends from the 25th to 75th percentile. The whiskers extend down to the 5% percentile value and up to the 95% percentile.

Rebuttal Figure 2

Rebuttal Figure 2: QQplot of the p-value distribution prior (x-axis) and after (y-axis) correcting for cell type composition.

The diagonal line denotes a perfect correlation of $-\log_{10}$ of the P -values before and after correction for cell content. The increase in significant P -values after correction for cell content indicates that the cell-type composition masked some of the significant findings. Examination of the P -value distribution (QQ-plot) did not show any evidence of a P -value inflation or any other strong unknown confounders associated with cell-type composition.

Rebuttal Figure 3

Rebuttal Figure 3: Expression of DLX1 and DLX1AS Transcripts

(a) Expression of *Dlx1* (blue) and *Dlx1as* (red), displayed as fragments per kilobase of transcript sequence per million mapped fragments (FPKM), in different cell types of mouse brain tissue [astrocytes (Astro), endothelial cells (Endo), newly formed oligodendrocytes (NFO), microglia (MGL), myelinating oligodendrocytes (MO), neuronal cells (Neuron), oligodendrocyte precursor cells (OPC); data from Zhang et al.²²] Both *Dlx1* and *Dlx1as* transcripts are almost exclusively expressed in neurons in the mouse brain.

(b) Total number of reads aligned to transcript sequences over all cells for the corresponding type of either *DLX1* (blue) or *DLX1AS* (red) from healthy human cerebral cortex (data from Darmanis et al.²³). Reads have been corrected for sequence depth of each cell and sequence length of each transcript and are

displayed as FPKM. Both *DLX1* and *DLX1AS* transcripts are almost exclusively expressed in neurons in the human cerebral cortex.

(c) Average number of reads per cell (mean) of either *DLX1* (blue) or *DLX1AS* (red) from healthy human cerebral cortex (Data from Darmanis et al.²³). Total number of reads of either *DLX1* or *DLX1AS* from cells of a specific type (e.g. neurons) have been divided by the total number of these cells. Reads have been corrected for sequence depth of each cell and sequence length of each transcript and are displayed as FPKM.

(d) N=130 individual neurons of healthy human cerebral cortex were tested for expression of *DLX1* or *DLX1AS*. This analysis showed that neurons either expressed *DLX1* (blue) or *DLX1AS* (red), or none of both (grey). Only 2.31% of neurons express both *DLX1* and *DLX1AS* (purple).

Reviewer #1 (Remarks to the Author):

The authors worked hard to answer the remaining question and I think they did a great job. Although I believe that more caution is warranted, the current dataset/analysis is probably the state of the field at this point. Future replications will eventually prove if the data is consistent. A few last remarks: Please include your power justification in the paper, I'm not sure why the authors calculate power for methylation differences <11% when most probes are around or less than 5% difference max. I guess the power looks much less convincing when the actual differences are used. Thank you for testing genotype effects using the Hannon data. I think it may nevertheless be useful to mention that so far unaccounted genetic variants potentially influence the result. This also applies to the cell type correction as current methods may not capture the full extend of confounding factors (cell types, genetics, other environmental data such as medication). A honest and comprehensive limitation senction would help to increase credibility of the paper. A last remark to the ~19,000 pages of supplement: Will there be a searchable electronic database ? Some annotation information for the CGs would be helpful (gene names? location?)

Reviewer #3 (Remarks to the Author):

The authors have addressed all of my concerns.

Although I was a little confused by the response by the authors regarding their post-hoc power analysis, the power for effect sizes > 10% change in DNAm levels (which I would argue is a large effect size for this measurement) seemed appropriate, and in line with previous calculations by our group for similar sample sizes.

Dear Dr. Trenkmann,

Thank you for your comments and the comments of reviewer #1 on our manuscript.

Following are revisions made and answers to your comments on the manuscript and the comments of reviewer #1.

Comment 1: Changes to Author list.

We have obtained agreement to the changes from all original authors. Their e-mails of consent are included.

On the title page the sentence “These authors jointly supervised to this work” had been included. We have deleted the “to”.

Comment 2: Please add at least one sentence that gives some background to the study...

We have modified the **Abstract** and start it with a more general statement:

“Genetic, epigenetic and environmental factors contribute to the multifactorial disorder Progressive supranuclear palsy (PSP). Investigation ...”

Comment 3: Please be more specific:

We are now more specific and write ... “DNA analysis from post-mortem tissue of forebrains of patients and controls.”

Comment 4: Could be left out to save space:

We also followed your suggestion and deleted the sentence “At ten genes methylation differences are ...” and eliminated the statement “These data are confirmed by siRNA mediated...” in order to stay within the word limits of the abstract.

Comment 5: *Please add a brief description of the main results of the study.*

We added a brief description of the results at the end of the **Introduction**:
“We describe significant DNA methylation differences between patients and controls at many CpG sites amounting to 451 protein coding genes. While methylation differences only affect one or a few sites at most genes, highly significant ($\geq 5\%$) hypermethylation is found at multiple sites associated with the gene *DLX1*. Functional analyses of both *DLX1* and its antisense transcript *DLX1AS* reveal an important role of *DLX1* in the pathogenesis of PSP.”

Comment 6: *Please note that large tables such as this must be provided as “Supplementary Data”*

Original Supplementary table 1 is now listed as Supplementary Data 1

Comment 7: *Provide all Supplementary Figures within a single merged Supplementary Information (SI) file...*

All Supplementary figures are now provided as a single merged Supplementary Information (SI) file. The cover page for the SI file containing the title of the ms and the name of the first author are given (SI file: Weber et al. “Epigenome-wide DNA methylation profiling in Progressive Supranuclear Palsy reveals major changes at *DLX1*”). We placed the Supplementary Figure legends directly with the respective item in the SI file.

We have added Supplementary Data 6 following the suggestion of reviewer #1. This table had been originally submitted as Rebuttal Table 1 and

contains data on the post-hoc power analysis. Furthermore we added three columns to this table that give chromosomal location and gene names of the CpGs as suggested by reviewer #1.

Comment 8: *Provide as high-resolution image ...*

We now provide a high-resolution image of Fig.2 (SVG vector file)

Comment 9: *Note that the Journal policy precludes “data not shown”*

In order to avoid “(data not shown)” we modified parts of the paragraph “**Single cell analysis in healthy human cortex**”. We write

DLX1 was expressed in 26.15%, DLX1AS in 15.39% and both were expressed in 2.31% of neurons (Supplementary Fig. 4d). The following sentences were deleted: “... as compared to 1.65% of all non-neuronal cells (not shown). The corresponding results for DLX1AS were 15.39% and 4.96% (not shown). While in most single neurons either DLX1, DLX1AS or none of both were detected, joint expression of DLX1 and DLX1AS was observed in 2.31%of neurons only.

Comments 10, 11: *Add size markers*

We have added size markers to fig. 4d and Supplementary Fig.5 and 6.

Comment 12: *Note that the journal policy precludes “data not shown”...*

We modified pertinent sentences of this paragraph:

“Initial experiments had shown that untreated Ntera2 cells express less DLX1 than SH-EP cells. DLX1 was overexpressed...”

We then proceeded

“...known target genes of DLX1, i.e. GAD1²⁴, GAD2²⁴, BRN3B²⁵, GnRH²⁶ and OLIG2²⁷. Of these genes GAD1, BRN3B, and OLIG2 were upregulated

in cells overexpressing *DLX1* (Fig. 5a) and downregulated in cells overexpressing *DLX1AS* (Fig. 5b). We ...“

modified the text as follows: Initial experiments ...than SH-EP cells.

Comment 13: *Did you confirm this by Western blot?*

Several Western blots showed that *DLX1* protein was reduced in cells treated with *DLX1*-targeting siRNAs as compared to controls. However, we did not perform extensive Western analyses as these data do not further contribute to the conclusions of the paper. Therefore we did not elaborate on Western analyses.

Comment 14: *...data not shown.*

See answer to comment 12

Comment 15: *Subheadings must be 60 characters or less.*

We shortened this subtitle to “siRNA-mediated down-regulation of *DLX1* and *DLX1AS*”

Comment 16: *Did you confirm this at the protein level?*

See answer to Comment 13.

Comment 17: *Provide this small table as ...*

We now provide Supplementary Table 1 within the merged SI file.

Comment 18: *... precludes “data not shown”.*

We deleted the sentence “In addition we also tested ... DLX1AS (data not shown)” since this observation is not essential for the conclusions of the paper.

Comment 19: ... elaborate whether these are existing ...”

Since we have deposited these accession numbers, we have moved this to “Data Availability”

Comments 20-22: Information about species and clones.

The informations are now given:

primary DLX1 antibody (dilution 1:1000, ab126054, polyclonal rabbit AB, Abcam, CA, UK)

Positive bands were detected using HRP-conjugated secondary antibody (dilution 1:5000, ab P0448, polyclonal goat anti rabbit immunoglobulins, affitinity isolated, Agilent, CA, USA)

β -actin antibody overnight (dilution 1:2000, ab 3700, clone 8H10D10, mouse monoclonal β -actin antibody, Cell Signaling Technology, CA, UK)

Comment 23: Update – if possible – references 58 and 64

References 58 and 64 were updated:

58. Fortin JP, Triche TJ Jr, Hansen KD. Preprocessing, normalization and integration of the Illumina HumanMethylationEPIC array with minfi. *Bioinformatics* 33, 558-560 (2017)

64. R Development Core Team R: A Language and Environment for Statistical Computing. Vienna, Austria: R Foundation for Statistical Computing (2011).

Comment 24: *Non-financial interests*

We add: The authors declare neither financial nor non-financial competing interests.

Comment 25: *n numbers, p values, name(s) of statistical tests, SD*

We carefully revised our figure legends for declarations of sample- or patient numbers ($n=...$), level of significance ($P=...$) and names of used statistical tests.

We add:

Figure 1

CpG sites hypo- and hypermethylated in $n=94$ PSP patients vs. $n=71$ controls are displayed on the left and right panels, respectively...

Figure 2

$N=451$ CpGs in protein coding genes and $n=26$ CpGs in non coding RNA genes were found differentially methylated. After removal of duplicates, i.e. genes with more than one differentially methylated CpG, $n=375$ genes show differential methylation in PSP patients.

The two outer circles list the autosomal positions of $n=375$ differentially methylated genes...

Comment 26: *Shorten figure title ...*

We shortened this figure title to “Overexpression and siRNA-mediated knock-down of *DLX1* and *DLX1AS*”

Comment 27: *Place directly with the respective item in the merged SI file...*

This has been done

Comment 28: *Provide legends for “Supplementary Data” in your cover letter, the legend for Supplementary Tables must be placed directly with the table in the merged SI file.*

Legends for the supplementary tables are now directly placed with the tables in the merged SI files

Reviewer #1 (Remarks to the Author):

The authors worked hard to answer the remaining question and I think they did a great job. Although I believe that more caution is warranted, the current dataset/analysis is probably the state of the field at this point. Future replications will eventually prove if the data is consistent. A few last remarks: Please include your power justification in the paper, I’m not sure why the authors calculate power for methylation differences <11% when most probes are around or less than 5% difference max. I guess the power looks much less convincing when the actual differences are used. Thank you for testing genotype effects using the Hannon data. I think it may nevertheless be useful to mention that so far unaccounted genetic variants potentially influence the result. This also applies to the cell type correction as current methods may not capture the full extend of confounding factors (cell types, genetics, other environmental data such as medication). A honest and comprehensive limitation senction would help to increase credibility of the paper. A last remark to the ~19,000 pages of supplement: Will there be a searchable electronic database ? Some annotation information for the CGs would be helpful (gene names? location?)

We originally referred to methylation differences of <11% based on the publication of Tsai et al. (2015). Tsai et al. had modeled ePower at different

methylation differences in order to estimate the power obtained by the analysis of cohorts of different sizes. This resulted in a power of 80% for the detection of methylation differences of < 11% in a study group of n=96 vs. a control group of n=96. (Tsai et al., *Int J Epidemiol.* 2015;44:1429-1441). We realize that mention of the <11% value is confusing in the context of power of various tests that we had calculated post-hoc. For this reason we removed mention of the <11% in the final version of the ms.

We included additional information on the post-hoc-power analysis (see **Methods** section, paragraph “**Array analysis**”):

“We computed the exact statistical power for each of the 485577 probe sets on the microarray based on the data obtained in our sample by applying the function `pwr.t2n.test` of the `pwr` package of R statistical software. The parameters of this function are the number of patients and controls, i.e. n=94 and n=71, respectively, as well as the significance level of 0.05 and Cohen’s d (effect size). The effect size is defined by the difference between the means of the group divided by the pooled standard deviations of the two groups. Given these parameters, a power of at least 80% is calculated for 14553 out of the total of 485577 probes. Of the 717 CpG sites which were significantly differentially methylated in PSP as compared to controls 664 (92.6%) showed a power of greater than 80% (Supplementary Data 6).”

Furthermore, table 1 of our previous rebuttal letter has been added as supplementary data 6. (see response to comment 7 and reviewer #1)

Reviewer #1 also asked for a statement on the limitations of our data. We added two sentences at the end of the second paragraph under the subheading “**Differentially methylated sites in PSP**” in the **Results** section:

“An influence of presently unknown genetic variants on the methylation pattern cannot be excluded. It is also not possible to correct for environmental factors such as individual medications and/or accompanying neurological diseases that might affect DNA methylation in the forebrain.”

We hope that our paper can now go online before too long and thank you again for your helpful comments on our manuscript.

Sincerely yours,

Ulrich Müller, Axel Weber, Günter Höglinger